# Fibroblast-enriched endoplasmic reticulum protein TXNDC5 promotes pulmonary fibrosis by augmenting TGFβ signaling through TGFBR1 stabilization

Tzu-Han Lee [1], Chih-Fan Yeh [1,2], Ying-Tung Lee[1], Ying-Chun Shih[1], Yen-Ting Chen[1], Chen-Ting Hung[1], Ming-Yi You[1], Pei-Chen Wu[1], Tzu-Pin Shentu[3], Ru-Ting Huang [3], Yu-Shan Lin[1], Yueh-Feng Wu[4], Sung-Jan Lin [4,5,6], Frank-Leigh Lu [7], Po-Nien Tsao [6,7], Tzu-Hung Lin[8], Shen-Chuan Lo[8], Yi-Shuan Tseng[1], Wan-Lin Wu[1], Chiung-Nien Chen[9], Chau-Chung Wu[2,10], Shuei-Liong Lin[6,11,12], Anne I. Sperling[3], Robert D. Guzy[3], Yun Fang [3] & Kai-Chien Yang [1,2,6,13]✉

Pulmonary fibrosis (PF) is a major public health problem with limited therapeutic options. There is a clear need to identify novel mediators of PF to develop effective therapeutics. Here we show that an ER protein disulfide isomerase, thioredoxin domain containing 5 (TXNDC5), is highly upregulated in the lung tissues from both patients with idiopathic pulmonary fibrosis and a mouse model of bleomycin (BLM)-induced PF. Global deletion of *Txndc5* markedly reduces the extent of PF and preserves lung function in mice following BLM treatment. Mechanistic investigations demonstrate that TXNDC5 promotes fibrogenesis by enhancing TGFβ1 signaling through direct binding with and stabilization of TGFBR1 in lung fibroblasts. Moreover, TGFβ1 stimulation is shown to upregulate TXNDC5 via ER stress/ATF6-dependent transcriptional control in lung fibroblasts. Inducing fibroblast-specific deletion of *Txndc5* mitigates the progression of BLM-induced PF and lung function deterioration. Targeting TXNDC5, therefore, could be a novel therapeutic approach against PF.

[1] Department and Graduate Institute of Pharmacology, National Taiwan University College of Medicine, Taipei, Taiwan. [2] Division of Cardiology, Department of Internal Medicine and Cardiovascular Center, National Taiwan University Hospital, Taipei, Taiwan. [3] Section of Pulmonary and Critical Care, Department of Medicine, University of Chicago, Chicago, IL, USA. [4] Department of Biomedical Engineering, College of Medicine and College of Engineering, National Taiwan University, Taipei, Taiwan. [5] Department of Dermatology, National Taiwan University Hospital and College of Medicine, Taipei, Taiwan. [6] Research Center for Developmental Biology & Regenerative Medicine, National Taiwan University, Taipei, Taiwan. [7] Department of Pediatrics, National Taiwan University Hospital, Taipei, Taiwan. [8] Material and Chemical Research Laboratories, Industrial Technology Research Institute, Zhudong, Taiwan. [9] Department of Surgery, National Taiwan University Hospital, Taipei, Taiwan. [10] Department and Graduate Institute of Medical Education & Bioethics, National Taiwan University College of Medicine, Taipei, Taiwan. [11] Department and Graduate Institute of Physiology, National Taiwan University College of Medicine, Taipei, Taiwan. [12] Division of Nephrology, Department of Internal Medicine, National Taiwan University Hospital, Taipei, Taiwan. [13] Institute of Biomedical Sciences, Academia Sinica, Taipei, Taiwan. ✉email: kcyang@ntu.edu.tw

Pulmonary fibrosis (PF), a condition of excessive scarring of the lungs, can be idiopathic (idiopathic pulmonary fibrosis, IPF) or secondary to various medical conditions. PF can lead to distorted pulmonary architecture, impaired lung function, and alveolar gas exchange, resulting in hypoxemia, dyspnea, exercise intolerance and ultimately, death. The global prevalence of IPF alone was estimated to be 5 million[1], and the total number of patients affected by PF is even higher due to numerous conditions that could lead to PF. Owing to the devastating morbidities/mortality and the tremendous amount of health care and economic burden associated with PF, it has become a major and growing public health problem. While the advances in medicine during the past few decades have improved dramatically the care of lung diseases, the incidence and mortality rate of PF, however, have barely improved[2,3]. A complex blend of cellular, molecular and structural changes may account for the progressive and intractable course of PF[2,3], and there is a clear and urgent need for novel therapies to improve the outcomes of PF patients.

With different underlying types of pathology, multiple mechanisms may account for the initiation and progression of PF. In general, alveolar epithelial injury resulting from exogenous (e.g., infection, toxin, radiation) or endogenous (eg. inflammation, oxidative stress, and aberrant immune responses) origins triggers the release of profibrotic cytokines (transforming growth factor-β [TGFβ] and tumor necrosis factor-α [TNFα]) and growth factors (connective tissue growth factor [CTGF], insulin-like growth factor 1[IGF-1] and platelet-derived growth factor [PDGF]). Increased local and circulating levels of profibrotic cytokines/growth factors stimulate the activation and proliferation of lung fibroblasts[4–6]. Activated lung fibroblasts transform into the α-smooth muscle actin (α-SMA)-expressing myofibroblasts, which are responsible for producing excessive amount of extracellular matrix (ECM) proteins that characterize the fibrotic lung tissue[7]. Myofibroblasts also modify ECM turnover by modulating the balance between matrix metalloproteinases (MMPs) and their natural inhibitors (tissue inhibitor of MMP; TIMPs) to promote PF[8]. Regardless of the initial trigger, unremitting fibroblast activation/myofibroblast differentiation and excessive accumulation of ECM play a central role in fibrosis formation and is observed in all types of PF.

Currently there are few clinical therapies directly targeting PF. Immunosuppressive and immunomodulatory therapies such as prednisone, azathioprine and N-acetyl-L-cysteine have been proven ineffective[9]. Monoclonal antibodies targeting TNFα (etanercept)[10] and CC-chemokine ligand 2 (CCL2, carlumab)[11] also failed to improve PF. Recently, pirfenidone (an inhibitor of multiple cytokines including TGFβ, PDGF, and TNFα)[12] and nintedanib (a tyrosine kinase inhibitor)[13] have been shown to be effective in slowing down the decline of lung function in IPF patients. However, neither agent stops the progression of PF, improves symptoms such as dyspnea and cough, or reduces the mortality of IPF. Moreover, the efficacy of either agent has yet to be determined in other forms of PF. There is, therefore, a clear and immediate need to identify additional novel mediators of PF to facilitate the development of useful, alternative therapeutic strategies targeting PF.

Recently, our group identified thioredoxin domain containing 5 (TXNDC5), a fibroblast-enriched endoplasmic reticulum (ER) protein, as a novel mediator of cardiac fibrosis[14]. Our studies suggest that TXNDC5 promotes cardiac fibrosis by redox-dependent cardiac fibroblast activation, as well as by enhancing extracellular matrix (ECM) production. In addition, we demonstrated that global deletion of Txndc5 protects against β agonist-induced cardiac fibrosis and contractile dysfunction in mice[14]. Because TXNDC5 is essential for the activation of cardiac fibroblasts and ECM protein production in response to injury, we

hypothesized that TXNDC5 could also play a critical role in activating fibroblasts in non-cardiac tissue, contributing to the development of organ fibrosis, including PF. We also engineered a new transgenic mouse line by which the causal role of fibroblast TXNDC5 in promoting PF can be determined in vivo.

In the present study, we demonstrated that TXNDC5 was highly upregulated in both human and mouse fibrotic lungs and that the expression level of TXNDC5 showed strong positive correlation with that of various fibrogenic protein genes. Results presented here suggest that TXNDC5 promotes PF by enhancing TGFβ signaling activity through post-translational stabilization and upregulation of TGFBR1 in lung fibroblasts, leading to increased fibroblast activation, proliferation, and excessive ECM production. In addition, we showed here that global deletion of Txndc5 protects against BLM-induced PF and lung dysfunction. Using a mouse line with inducible, fibroblast-specific Txndc5 deletion, we have further demonstrated that inducing Txndc5 deletion in lung fibroblasts lessened the development and progression of lung fibrosis and lung function decline in mice with BLM-induced PF. Taken together, these results uncovered a previously unrecognized role of ER protein TXNDC5 in modulating TGFβ signaling activity and in the development of lung fibrosis. Our data also suggest that targeting TXNDC5 could be a powerful new therapeutic approach to mitigate lung fibrosis, thereby improving lung function and outcomes in patients with PF.

## Results

**TXNDC5 was upregulated in human IPF lungs/lung fibroblasts.** To determine the potential role of TXNDC5 in PF, we analyzed the transcript and protein expression levels of TXNDC5 in the lung tissues from control subjects and IPF patients. As shown in Fig. 1a, the expression of TXNDC5 mRNA was markedly increased in human IPF, compared with control, lung tissues. In addition, using fresh protein extracts of the lung tissues from control subjects and IPF patients, immunoblots showed that the protein expression levels of TXNDC5, as well as the activated myofibroblast marker αSMA, were significantly increased in human IPF, compared with control, lungs (Fig. 1b). Consistent with these findings, re-analyses of multiple microarray datasets obtained from human lung tissues (GSE72073, Supplementary Fig. 1a) and lung fibroblasts (GSE40839, Supplementary Fig. 1b)[15] of control and IPF patients also showed marked upregulation of TXNDC5 mRNA in the lung tissues/lung fibroblasts from IPF patients, compared with that from control subjects. Moreover, immunoblots using protein lysates of lung fibroblasts from IPF patients showed marked upregulation of TXNDC5 and αSMA proteins, compared with that from control subjects, in response to TGFβ1 treatment (Fig. 1c). Similar to these results, analyses of these microarray data also revealed a strong positive correlation between the expression levels of TXNDC5 and that of multiple ECM protein genes such as COL1A1, ELN, and myofibroblast activation marker ACTA2 (which encodes αSMA) in human lung fibroblasts (Fig. 1d). Collectively, increased expression of TXNDC5 transcript and protein in human IPF lungs and lung fibroblasts, as well as its strong positive correlation with multiple fibrogenic protein genes suggest the potential importance of TXNDC5 in the pathogenesis of lung fibrosis.

**TXNDC5 was upregulated in BLM-induced fibrotic mouse lungs.** To determine the in vivo functional role of TXNDC5, we used a mouse model of lung fibrosis induced by intra-tracheal instillation of bleomycin (BLM, 3 mg/kg). Consistent with the induction of lung fibrosis by BLM, the transcript expression levels of various fibrogenic protein genes including Col1a1, Col3a1, Eln,

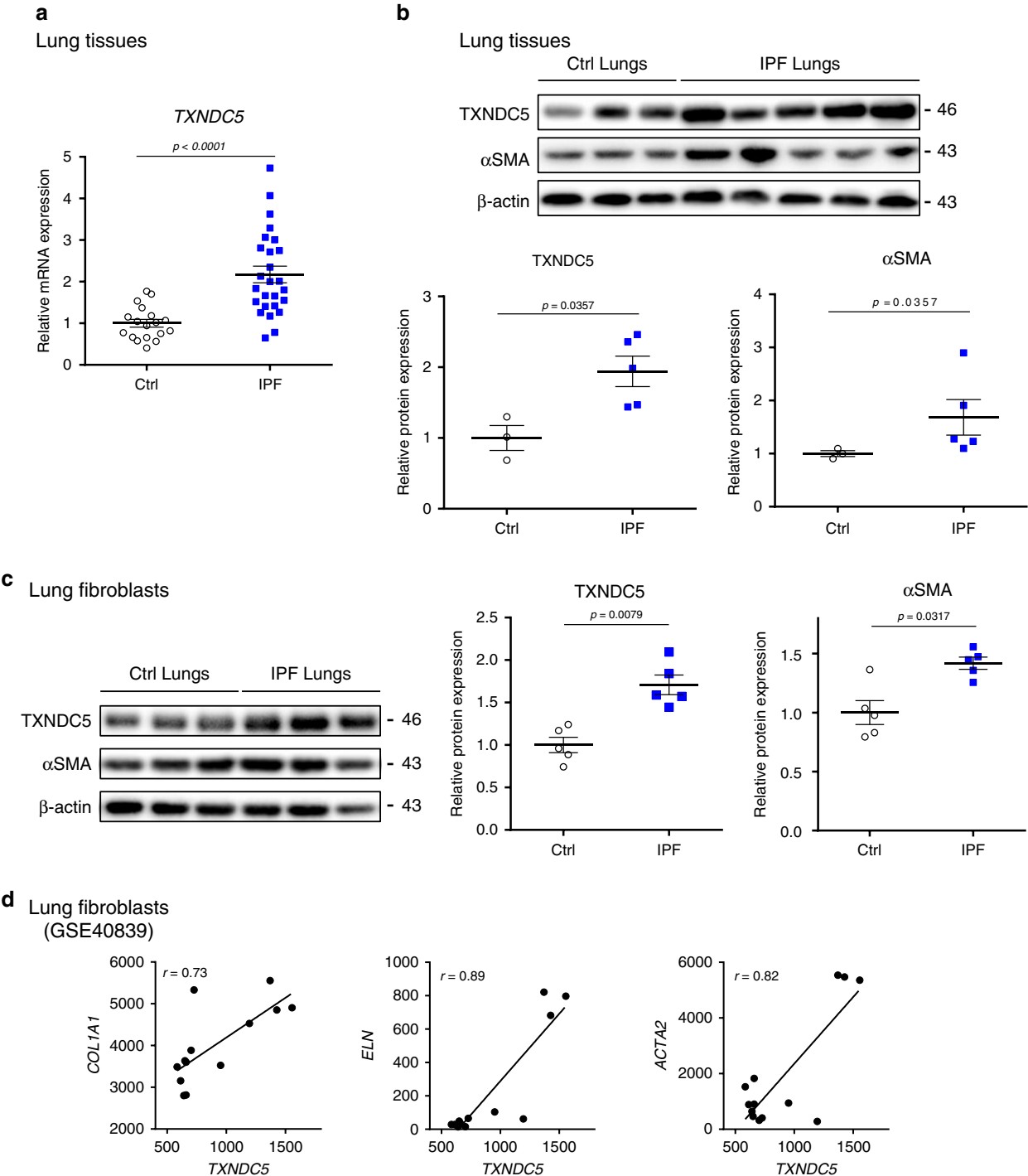

**Fig. 1 TXNDC5 was upregulated in IPF lungs/lung fibroblasts and correlated with fibrogenic genes. a** Transcript analysis showed significant upregulation of *TXNDC5* mRNA in human IPF ($n = 26$, central and lateral lung from 13 biologically independent samples), compared with control ($n = 18$, central and lateral lung from 9 biologically independent samples), lung tissues. **b** Immunoblot analysis showed a marked increase in TXNDC5 and αSMA protein levels in human IPF ($n = 5$ biologically independent samples), compared with control ($n = 3$ biologically independent samples), lung tissues. **c** Protein expression level of TXNDC5 and αSMA were also increased in IPF, compared with control, lung fibroblasts following TGFβ1 treatment ($n = 5$ biologically independent samples per group). **d** Re-analyses of microarray data from IPF human lung fibroblasts (GSE40839)[15] revealed strong positive correlation between the expression level of *TXNDC5* and that of genes encoding fibrogenic proteins (*COL1A1, ELN, ACTA2*) in human lung fibroblasts ($n = 13$ biologically independent samples per group) (Data are presented as mean ± SEM, *P* value determined using two-tailed Mann–Whitney *U* test. Source data are provided as a Source Data file).

*Fn, Ctgf*, as well as *Txndc5*, were significantly upregulated in BLM-treated mouse lungs (21 days post-BLM treatment, Fig. 2a). In addition, immunohistochemical (IHC) staining revealed distorted alveolar structure, thickened alveolar walls, and formation of fibrotic foci with strong staining of αSMA in BLM-treated mouse lungs (Fig. 2b, inset). IHC staining in serial sections of fibrotic mouse lungs also detected marked increases in TXNDC5 staining, particularly in the αSMA-positive fibrotic foci (Fig. 2b, inset).

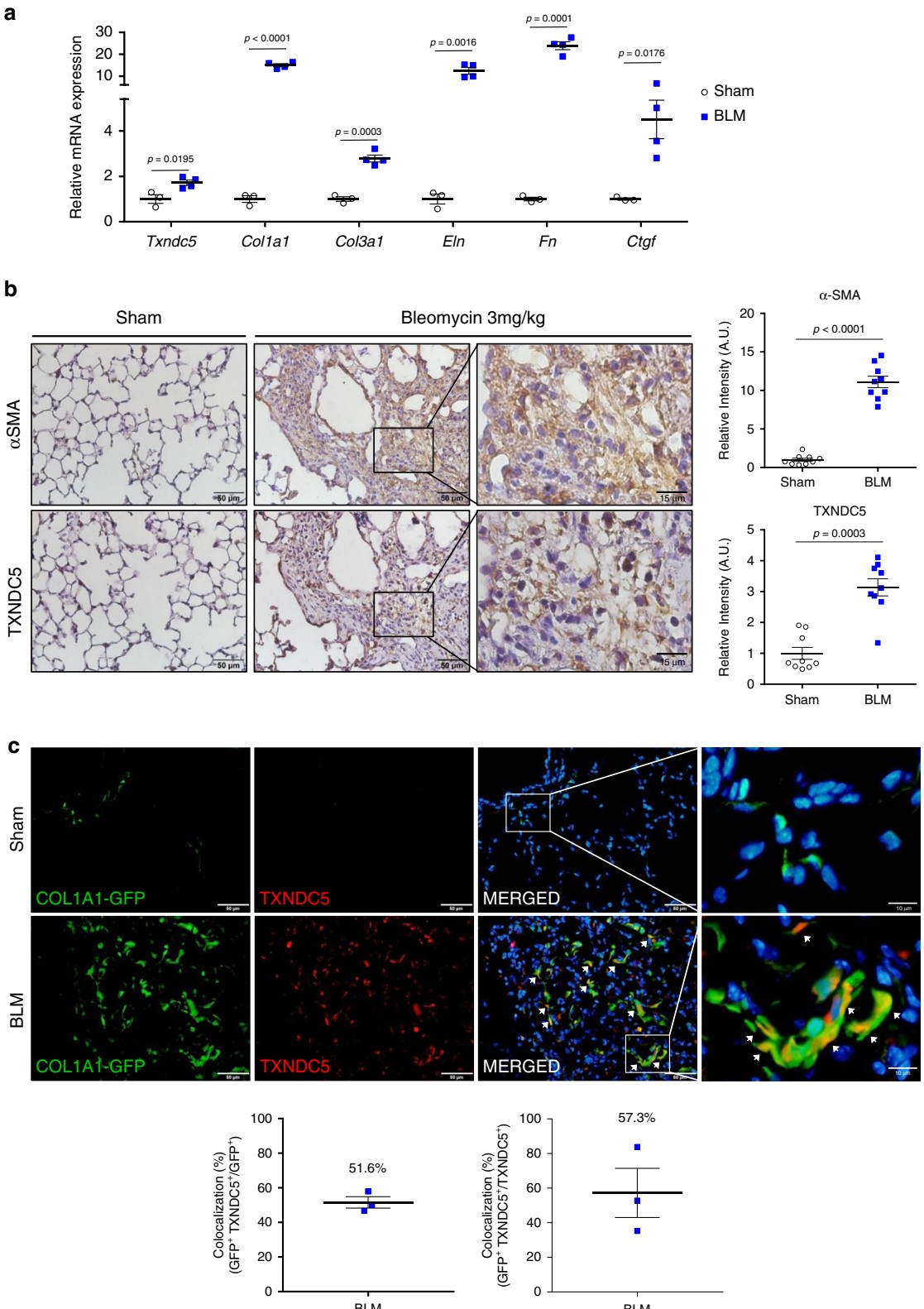

Using a *Col1a1-GFP*^Tg (GFP driven by *Col1a1* enhancer/promoter) transgenic mouse line that allows visualizing active lung fibroblasts with GFP[16], immunofluorescence (IF) staining showed that TXNDC5 staining was highly co-localized with GFP-positive, collagen-producing lung fibroblasts (Fig. 2c, arrows and inset) and activated myofibroblasts (α-SMA positive, Supplementary Fig. 2a and b, arrows and inset) in fibrotic mouse lungs

following BLM treatment. IF staining using the lung sections from a BLM-treated, tamoxifen-inducible transgenic mouse line *Tie2-Cre/ERT2*ROSA26-tdTomato* (*Tie2-tdTomato*) that allows the visualization of endothelial cells with red fluorescence protein tdTomato, showed that TXNDC5 was only expressed in ~37% of tdTomato-positive lung endothelial cells (Supplementary Fig. 2c, d). In addition, IF staining showed that TXNDC5 did not

**Fig. 2 TXNDC5 was upregulated in BLM-induced PF and enriched in lung fibroblasts in vivo. a** Transcript expression levels of *Txndc5* and fibrogenic protein genes (*Col1a1, Col3a1, Eln, Fn* and *Ctgf*) were markedly increased in the lung tissues from BLM-(*n* = 4 biologically independent animals per group), compared with PBS- (Sham, *n* = 3 biologically independent animals), treated WT mice on Day 21. **b** Immunohistochemical (IHC) staining on the serial sections of mouse lungs showed a strong upregulation of αSMA and TXNDC5 in the lung tissues from BLM-, compared with PBS- (Sham), treated WT mice on Day 21 (*n* = 9 fields examined over 3 biologically independent animals per group). **c** IF staining for TXNDC5 on lung sections from sham-operated and BLM-treated *Col1a1-GFP*^Tg^ mice. BLM treatment significantly increased TXNDC5 expression and the number of GFP-positive cells in the mouse lungs on Day 21 (*n* = 3 biologically independent animals). There was a high degree of co-localization of TXNDC5 with GFP-positive, collagen producing lung fibroblasts (white arrows and inset) (data are presented as mean SEM, *P* value determined using two-tailed Mann–Whitney *U* test. Source data are provided as a Source Data file. BLM bleomycin).

co-localized with T1α, a marker for type I pneumocytes, in BLM-treated mouse lungs (Supplementary Fig. 2e). In BLM-treated *Col1a1-GFP*^Tg^ mouse lungs, the staining of surfactant protein-C (SP-C), a marker for type II pneumocytes, was mutually exclusive from GFP-positive, collagen-secreting cells, suggesting that type II pneumocytes do not express collagen and may not express appreciable level of TXNDC5 (Supplementary Fig. 2f). Consistent with this result, IF staining using the lung sections from a BLM-treated, tamoxifen-inducible transgenic mouse line *SPC-Cre/ERT2*ROSA26-mTmG* (*SPC-mTmG*) that allows visualizing type II pneumocytes with GFP following tamoxifen injection, showed that TXNDC5 was not expressed at appreciable levels in type II pneumocytes (Supplementary Fig. 2g). Taken together, these data revealed fibroblast-specific expression and upregulation of TXNDC5 in fibrotic mouse lungs that correlate well with ECM gene expression and lung fibroblast activation, strongly suggesting its role in regulating lung fibroblast function and the formation of PF.

**Global deletion of *Txndc5* attenuated PF induced by BLM.** To determine the requirement of TXNDC5 in the development of lung fibrosis, WT, and *Txndc5*^−/−^ mice[14] were subjected to BLM treatment (3 mg/kg, intratracheal instillation). As shown in Fig. 3a–c, microCT scanning showed that BLM treatment led to marked lung destruction, volume reduction, and extensive lung fibrosis in WT mice at 21 days, whereas knocking out *Txndc5* preserved the lung structure and attenuated the volume reduction and fibrotic changes induced by BLM treatment. Consistent with these results, picrosirius red (Fig. 3d) and Masson's trichrome staining (Supplementary Fig. 3) showed that global deletion of *Txndc5* markedly reduced the extent of pulmonary fibrosis following BLM treatment. In addition, BLM treatment resulted in a marked increase in the number of GPF-positive, collagen-producing lung fibroblasts in the lung tissue from *Col1a1-GFP*^Tg^ reporter mice, whereas deletion of *Txndc5* in *Col1a1-GFP*^Tg^ mice (by crossbreeding *Txndc5*^−/−^ with *Col1a1-GFP*^Tg^ mice) significantly reduced the number of GFP-positive lung fibroblasts following BLM treatment (Fig. 3d). As shown in Fig. 3e, hydroxyproline content, a biochemical index for pulmonary fibrosis and collagen deposition, was markedly increased in WT, but not in *Txndc5*^−/−^, mouse lungs in response to BLM treatment. Functionally, pressure-volume (PV) curves in BLM-challenged mice assessed using FlexiVent FX system (SCIREQ Inc, Montreal, Canada) showed a right-downward shift, corresponding to a decrease of lung compliance compared to sham-operated animals (Fig. 3f, h). The PV curve, however, was less shifted in *Txndc5*^−/−^ mice following BLM treatment (Fig. 3f). Calculation from the FlexiVent snapshot perturbation showed that BLM treatment led to a significant increase in airway resistance and lung elastance, as well as a reduction in lung compliance and inspiratory capacity in WT mice, consistent with the development of lung fibrosis. All these lung functional abnormalities caused by BLM treatment were significantly attenuated in *Txndc5*^−/−^ mice (Fig. 3g, h). Taken together, these data showed

that BLM treatment led to marked lung fibrosis and lung function impairment, whereas global deletion of *Txndc5* protected against BLM-induced lung fibrosis and preserved lung function. These results strongly suggest the central role of TXNDC5 in the pathogenesis of PF.

**Txndc5 deletion did not alter inflammatory response to BLM.** Because inflammatory response is a crucial mediator of lung fibrosis following BLM treatment, we went on to examine if global deletion of *Txndc5* would impact pulmonary inflammation induced by BLM. As shown in Fig. 4a (D14 post-BLM) and Supplementary Fig. 4a (D7 post-BLM), the bronchoalveolar lavage fluid (BALF) from WT mice 1-2 weeks following BLM treatment showed markedly increased total protein content, and modified Giemsa stain of BALF revealed increased alveolar infiltration of inflammatory cells including macrophages, neutrophil, and lymphocytes. The number of inflammatory cells in the BALF from BLM-treated *Txndc5*^−/−^ mice, however, was similar to that observed in WT mice. Consistent with this finding, the expression levels of inflammatory cytokines including *Il6* and *Il1b* were similarly elevated in BLM-treated WT and *Txndc5*^−/−^ mouse lungs (Fig. 4b). In addition, immunofluorescence (IF) staining showed that TXNDC5 did not co-localized with CD11b, a marker for inflammatory cells (Fig. 4c). Taken together, these data suggest that TXNDC5 is not required for development of pulmonary inflammation in response to BLM treatment.

**TXNDC5 is causative in triggering fibrogenecity in HPF.** To determine the functional role of TXNDC5 in lung fibroblasts, *TXNDC5* was knocked down in human pulmonary fibroblasts (HPF) with shRNA, followed by TGFβ1 stimulation. As shown in Fig. 5a, b, *TXNDC5* knock-down (KD) significantly reduced ECM (COL1A1, fibronectin, and elastin), αSMA/*ACTA2* and periostin (markers for lung fibroblast activation) protein/gene expression induced by TGFβ1 stimulation in HPF. Consistent with reduced lung fibroblast activity (reflected in reduced αSMA/periostin expression) with *TXNDC5* KD, the cellular proliferation induced by TGFβ1 was significantly attenuated by TXNDC5 depletion (Fig. 5c). Similar to the results in HPF, fibrogenic proteins (COL1A1 and αSMA) and cellular proliferation activity were increased in WT, but not in *Txndc5*^−/−^, isolated mouse lung fibroblasts (MLF, Supplementary Fig. 5a, b) following TGFβ1 stimulation. These data collectively suggest that TXNDC is required for TGFβ1-induced lung fibroblast activation, proliferation, and ECM production.

Next, we went on to determine if increased TXNDC5 expression *per se* could alter the cellular activity and ECM production in human lung fibroblasts. As shown in Fig. 5d, forced expression of *TXNDC5* in HPF led to strong upregulation of ECM proteins (fibronectin and COL1A1) and markers for fibroblast activation (periostin and α-SMA), compared to empty vector-transduced HPF. *TXNDC5* overexpression (OE) also led to significantly increased proliferative activity in human lung fibroblasts (Fig. 5e). These data show that increased TXNDC5

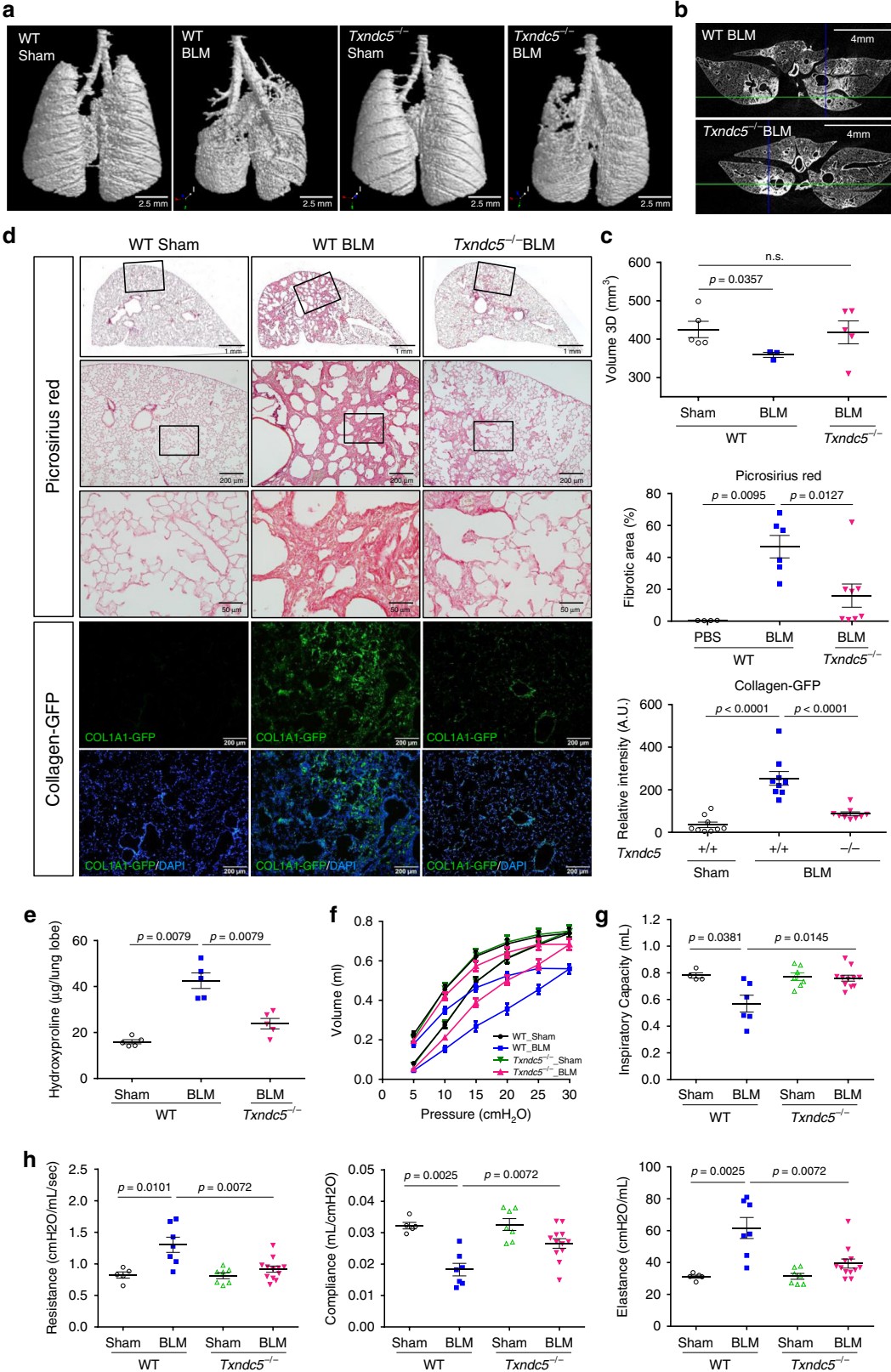

expression is sufficient to trigger human lung fibroblast activation, proliferation, and ECM protein production.

**TXNDC5 augments TGFβ signaling activity by increasing TGFBR1.** Because TXNDC5 is required for TGFβ1-induced HPF activation (Fig. 5), we hypothesized that TXNDC5 may regulate HPF activity through effectors downstream of the TGFβ1 signaling pathway. Indeed, we detected that knockdown of *TXNDC5* in HPF significantly reduced TGFβ1-induced phosphorylation of SMAD3 (canonical TGFβ1 signaling,

**Fig. 3 Knockout of *Txndc5* markedly reduced the severity of pulmonary fibrosis induced by BLM. a, b** Representative microCT images of lung tissues from WT and *Txndc5*$^{-/-}$ mice with and without BLM treatment on Day 21. MicroCT imaging showed marked parenchymal destruction and volume reduction in WT mouse lungs following BLM treatment (**a**). Lung volume reduction and fibrotic changes (white areas in **b**) were significantly attenuated in *Txndc5*$^{-/-}$ mice. **c** Quantification of lung volume by microCT in WT and *Txndc5*$^{-/-}$ mice with and without BLM treatment on Day 21 (WT sham $n = 5$, WT BLM $n = 3$, *Txndc5*$^{-/-}$ BLM $n = 5$ biologically independent animals). **d** Picrosirius red staining (top panels) of lung sections from WT and *Txndc5*$^{-/-}$ mice 21 day after intra-tracheal administration of BLM. Images in the lower panels were magnified from the inset of the photomicrographs in the upper panels. Middle panel showed the representative fluorescence images of lung sections from *Col1a1-GFP*$^{Tg}$ and *Col1a1-GFP*$^{Tg*}$*Txndc5*$^{-/-}$ mice treated with BLM or PBS (Sham). Quantitative results of picrosirius red- (WT sham $n = 4$, WT BLM $n = 6$, *Txndc5*$^{-/-}$ BLM $n = 8$ biologically independent animals) and GFP-positive areas ($n = 9$ fields examined over 3 biologically independent animals per group) determined from above were shown on the right. **e** Hydroxyproline content was markedly increased in the mouse lungs from WT mice treated with BLM on Day 21. Global deletion of *Txndc5* significantly reduced hydroxyproline content in the mouse lungs following BLM treatment ($n = 5$ biologically independent animals per group). **f** Representative pressure-volume curves (WT sham $n = 4$, WT BLM $n = 7$, *Txndc5*$^{-/-}$ BLM $n = 7$, *Txndc5*$^{-/-}$ BLM $n = 11$ biologically independent animals) and (**g, h**) lung function parameters determined using flexiVent FX system in WT and *Txndc5*$^{-/-}$ mice 21 days following sham procedure or BLM treatment(g: WT sham $n = 4$, WT BLM $n = 6$, *Txndc5*$^{-/-}$ BLM $n = 7$, *Txndc5*$^{-/-}$ BLM $n = 11$ biologically independent animals, h: WT sham $n = 5$, WT BLM $n = 7$, *Txndc5*$^{-/-}$ BLM $n = 7$, *Txndc5*$^{-/-}$ BLM $n = 12$ biologically independent animals) (Data are presented as mean ± SEM, $P$ value determined using two-tailed Mann–Whitney $U$ test. Source data are provided as a Source Data file. n.s. non-significant, BLM bleomycin).

Supplementary Fig. 6a, b), JNK and ERK (non-canonical TGFβ1 signaling, Supplementary Fig. 6a, c), whereas TXNDC5 overexpression in HPF was sufficient to trigger the activation of SMAD3, JNK, and ERK (Supplementary Fig. 6d, e).

Because the canonical (SMAD3) and non-canonical (JNK and ERK) TGFβ1 signaling pathways were both affected by TXNDC5, we hypothesized that TXNDC5 could modulate TGFβ1 signaling activity upstream of these kinase effectors, likely at the ligand/ receptor level. Consistent with this hypothesis, TGFβ1 treatment induced marked upregulation of TGFβ receptor 1 (TGFBR1) protein in HPF, which was abolished by *TXNDC5* knockdown (Supplementary Fig. 7 and Fig. 6a). By contrast, *TXNDC5* knockdown did not affect TGFβ receptor 2 (TGFBR2) expression in HPF (Fig. 6a). In addition, forced expression of *TXNDC5* increased TGFBR1, but not TGFBR2 expression levels in HPF (Fig. 6b). Consistent with these in vitro findings, in vivo studies also showed that TGFBR1 was significantly upregulated in BLM-treated WT, but not in *Txndc5*$^{-/-}$, mouse lungs (Fig. 6c). In addition, IF staining of the lung sections from BLM-treated and sham-operated *Col1a1-GFP*$^{Tg}$ mice showed that TGFBR1 was markedly upregulated in BLM-treated mouse lungs, especially in GFP$^+$ collagen-secreting lung fibroblasts (Fig. 6d). Global deletion of *Txndc5*, however, prevented the upregulation of TGFBR1 in *Col1a1-GFP*$^{Tg}$ mouse lungs in response to BLM treatment (Fig. 6d, e). In addition, treatment with LY364947 (10 μM), a TGFβ receptor 1 kinase inhibitor, abolished the activation of SMAD and prevented the upregulation of COL1A1, fibronectin, and periostin (Fig. 7a) induced by TXNDC5 over-expression in HPF. Moreover, shRNA targeting TGFBR1 (sh*TGFBR1*) abrogated TXNDC5 overexpression-induced HPF activation and ECM protein production. (Fig. 7b). Collectively, these data demonstrate that the profibrogenic effects of TXNDC5 were mediated through increased TGFβ1 signaling activity via increased TGFBR1 expression levels.

**TXNDC5 stabilizes TGFBR1 protein by facilitating its folding.** To determine how TXNDC5 regulates TGFBR1 protein expression in HPF, a cycloheximide protein chase assay was conducted to assess the protein stability of TGFBR1 in HPF with *TXNDC5* knockdown or overexpression. As shown in Fig. 8a, knockdown of *TXNDC5* in HPF led to accelerated degradation of TGFBR1, compared to shScr-treated cells. On the other hand, over-expression of *TXNDC5* increased TGFBR1 protein stability, compared to empty vector-transduced HPF (Fig. 8b). These data suggest that TXNDC5 enhances TGFBR1 expression through, at least partially, stabilizing TGFBR1 protein.

TXNDC5, an ER resident protein, is a member of protein disulfide isomerase (PDI) family with thioredoxin activity known to catalyze the rearrangement of disulfide bonds, suggesting that TXNDC5 could facilitate protein folding in the ER. It is known that integral membrane and secreted proteins fold in the ER by chaperons or PDIs prior to membrane transit to cell surface[17,18]; misfolded proteins, however, are subjected to proteasome-mediated degradation[19]. We hypothesized that TXNDC5 facilitates the folding of TGFBR1, allowing the maturation and transit of TGFBR1 to the plasma membrane. Loss of TXNDC5, therefore, could lead to TGFBR1 misfolding and subsequent degradation mediated by proteasome. Consistent with this hypothesis, in situ proximity ligation assays (PLA) showed that TXNDC5 interacts with TGFBR1 (Supplementary Fig. 8a), but not TGFBR2 (Supplementary Fig. 8b), in HPF. In addition, treatment with proteasomal inhibitor MG132 (20 μM) partially restored TGFBR1 protein expression in lung fibroblast with *TXNDC5* knockdown (Fig. 8c).

Importantly, overexpression of a mutant TXNDC5 that lacks PDI activity (TXNDC5-AAA mutant: cysteine-to-alanine mutations were introduced in both ends of each of its 3 thioredoxin domains [CGHC to AGHA], thereby abolishing its PDI activity, scheme shown in Supplementary Fig. 8d)[14] failed to increase the expression levels of TGFBR1 in HPF (Fig. 8e), suggesting that TXNDC5-mediated upregulation of TGFBR1 is dependent on its PDI activity. TXNDC5-AAA mutant also failed to increase the expression levels of ECM (COL1A1 and fibronectin) proteins and fibroblast activation markers (periostin and αSMA, Fig. 8e). Consistent with these observations, additional co-immunoprecipitation experiments showed that TGFBR1 interacts with TXNDC5 (Supplementary Fig. 8c), but not with TXNDC5-AAA mutant (Supplementary Fig. 8d). Taken together, these results demonstrate that TXNDC5 functions to maintain the stability of TGBFR1 protein, likely by facilitating the proper folding of TGFBR1 protein through its PDI activity, in lung fibroblasts. When TXNDC5 expression is reduced in lung fibroblasts, TGFBR1 becomes unfolded and subsequently degraded by proteasome, thereby leading to reduced TGFβ signaling activity, ECM production, and fibroblast proliferation/ activation.

**TGFβ1 induces TXNDC5 via ATF6 ER stress pathway in HPF.** Next, we went on to investigate how TXNDC5 is regulated by TGFβ1 in lung fibroblasts. Because the transcript expression level of *TXNDC5* is upregulated in response to TGFβ1 stimulation, we hypothesized that *TXNDC5* promoter is regulated by transcription factor(s) downstream of TGFβ1 signaling pathway. In our

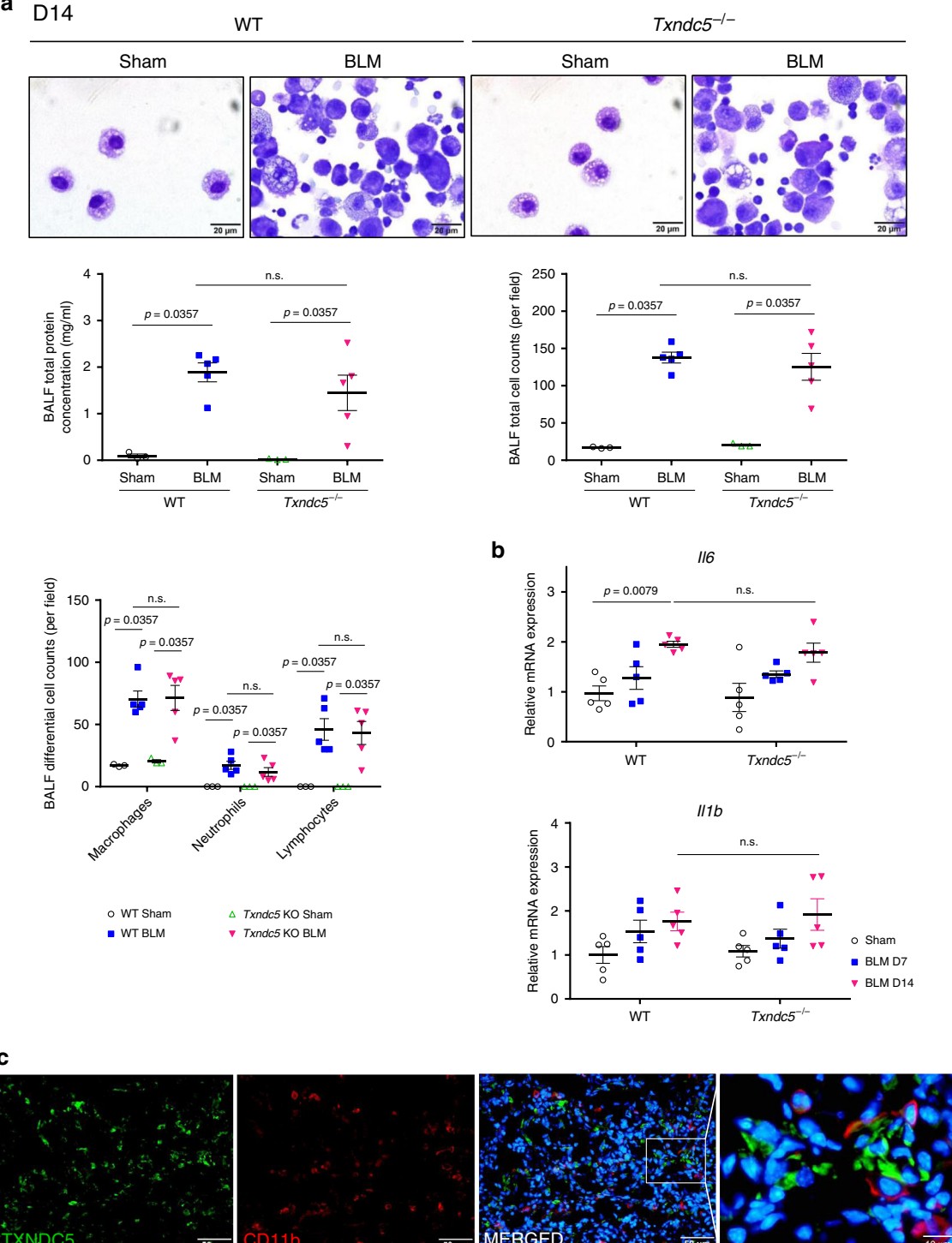

**Fig. 4 Global deletion of *Txndc5* did not alter inflammatory response to BLM treatment. a** Representative photomicrographs of modified Giemsa staining of the bronchoalveolar lavage fluid (BALF) from WT and *Txndc5*−/− mice 14 days after intra-tracheal BLM or PBS (Sham) instillation (top panel). Quantification of total protein content and number of inflammatory cell (macrophage, neutrophil and lymphocytes) in the BALF from each group of the experimental animals (bottom panel) (WT sham n = 3, WT BLM n = 5, *Txndc5*−/− sham n = 3, *Txndc5*−/− BLM n = 5 biologically independent animals). **b** Transcript expression levels of pro-inflammatory cytokines, *Il6* and *Il1b*, quantified in the lung tissues from WT and *Txndc5*−/− mice 7 days and 14 days after BLM or PBS (Sham) treatment (n = 5 biologically independent animals per group). **c** Immunofluorescence staining of TXNDC5 (green) and CD11b (red) of lung sections from WT mice 14 days following BLM treatment. TXNDC5 was not present in CD11b-positive, inflammatory cells (Data are presented as mean ± SEM, *P* value determined using two-tailed Mann–Whitney *U* test. Source data are provided as a Source Data file. n.s. non-significant, BLM bleomycin).

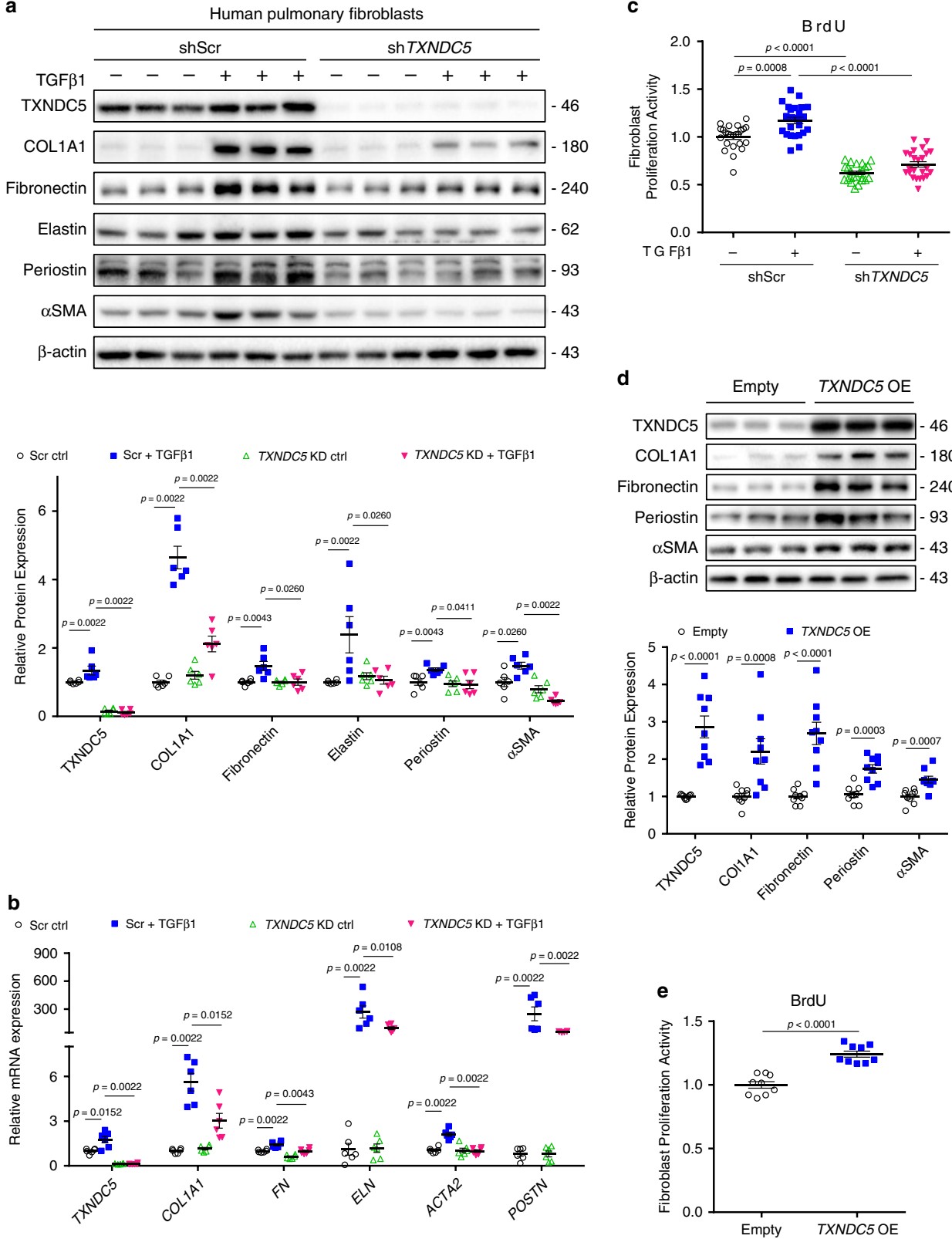

previous studies, we have demonstrated that in cardiac fibroblasts, TGFβ1 upregulates TXNDC5 through ER stress pathway activation and ATF6-dependent transcriptional control[14]. In line with these findings, TGFβ1 treatment in HPF markedly increased ER stress signaling activity, as evidenced by upregulated ER stress markers including BiP and components of all three branches of

ER stress signaling including ATF6 (Fig. 9a), inositol-requiring enzyme 1 (IRE1), and protein kinase RNA-like endoplasmic reticulum kinase (PERK) pathways (Supplementary Fig. 9a, b, ER stress inducer tunicamycin used as a positive control). TGFβ1-induced increases in ER stress markers (BiP and ATF6α(N), Fig. 9a), as well as transcriptional upregulation of *TXNDC5* in

**Fig. 5 TXNDC5 is both essential and sufficient for HPF activation, proliferation, and ECM production.** Protein ($n = 6$ biologically independent samples per group) (**a**) and transcript ($n = 6$ biologically independent samples per group) (**b**) expression levels of COL1A1, fibronectin, elastin, periostin, and αSMA/*ACTA2* were markedly increased in control (shScr) human pulmonary fibroblasts (HPF) following TGFβ1 (10 ng/ml) treatment. TGFβ1-induced upregulation of these fibrogenic proteins/genes was significantly attenuated in HPF with TXNDC5 depletion (sh*TXNDC5*). **c** TGFβ1 treatment significantly increased the cellular proliferation activity in shScr-, but not in sh*TXNDC5*-, transduced HPF ($n = 24$ biologically independent samples per group). ECM proteins (COL1A1 and fibronectin), markers for fibroblast activation (periostin and αSMA) ($n = 9$ biologically independent samples per group) (**d**) and fibroblast proliferation activity ($n = 9$ biologically independent samples per group) (**e**) were markedly increased in HPF transduced with *TXNDC5* (TXNDC5 OE), compared with empty, expression vector (Data are presented as mean ± SEM, *P* value determined using two-tailed Mann–Whitney *U* test. Source data are provided as a Source Data file. n.s. non-significant, ctrl control, KD knockdown, OE overexpress).

HPF (Fig. 9b), were attenuated with the treatment with ER stress inhibitor 4-phenylbutyrate (4-PBA, 2 mM; Fig. 9a, b). These data suggest that TGFβ1-induced upregulation of *TXNDC5* in HPF is dependent on increased ER stress levels.

To determine which ER stress pathway controls the transcriptional regulation of *TXNDC5*, *ATF6*, *XBP-1* and *EIF2A*, the essential components downstream of ATF6, IRE1 and PERK pathways, respectively, were knocked down in HPF (Fig. 9c and Supplementary Fig. 10a, c) followed by TGFβ1 treatment. As shown in Fig. 9d, TGFβ1-induced *TXNDC5* upregulation in HPF was completely blocked by *ATF6* depletion, but not by knockdown of *XBP1* or *EIF2A* (Supplementary Fig. 10b, d). In addition, wild-type (WT *TXNDC5*) and ATF6 binding site (TGACGTGG, +642~+653)-deleted (ΔATF6) human *TXNDC5* promoter-luciferase constructs were transfected into HPF, followed by TGFβ1 stimulation. As shown in Fig. 9e, TGFβ1 significantly increased WT *TXNDC5* promoter activity, whereas deletion of the ATF6 binding site markedly reduced *TXNDC5* promoter activity at baseline and in response to TGFβ1 stimulation. Taken together, these data demonstrate that TGFβ1-induced TXNDC5 upregulation in HPF is dependent on increased ER stress and ATF6-mediated transcriptional regulation.

**Fibroblast-specific *Txndc5* deletion lessened BLM-induced PF.** To determine the in vivo contribution of fibroblast TXNDC5 to pulmonary fibrosis, we generated a mouse line with inducible, fibroblast-specific *Txndc5* deletion (*Col1a2-Cre/ERT2\*Txndc5^fl/fl*, abbreviated as *Txndc5^cKO*, Supplementary Fig. 11). Tamoxifen (80 mg/kg i.p. every other day) was administered in *Txndc5^cKO* mice 7 days after BLM instillation, a time point where marked PF was observed in the mouse lungs (Fig. 10a), followed by assays to quantify pulmonary function and extent of lung fibrosis at 21 days post-BLM treatment. In these experiments, tamoxifen-treated *Col1a2-Cre/ERT2* (*Col1a2-Cre*) mice were used as controls. As shown in Fig. 10b, c, e, marked lung fibrosis, as evidenced by increased picrosirius red staining area and elevated hydroxyproline content, was established both in control and *Col1a2-Cre/ERT2\*Txndc5^fl/f* mice 7 days after BLM treatment. Targeted deletion of *Txndc5* in lung fibroblasts by tamoxifen injection in *Col1a2-Cre/ERT2\*Txndc5^fl/fl* mice since Day 7 after BLM instillation (the efficiency of TXNDC5/*Txndc5* reduction in fibrotic mouse lungs was shown in Supplementary Fig. 12a, b), however, significantly attenuated the expansion of picrosirius red-positive areas and the increases in hydroxyproline content on Day 21 post-BLM treatment, suggesting a lessened degree of PF progression, comparing to BLM-treated control mice. Consistent with these findings, second harmonic generation (SHG) microscopy, an imaging approach widely employed to image fibrillar collagen, showed rapid progression of PF in control mouse lungs between day 7 and 21 after BLM instillation (Fig. 10b, d). The progression of lung fibrosis was halted in *Txndc5^cKO* mice on SHG imaging. In addition, the induction of *Txndc5* deletion in fibroblasts also prevented or partially reversed lung function decline (parameters including PV curve, resistance, compliance,

and elastance, Fig. 10f, g) in BLM-treated mice. BLM treatment led to marked upregulation of ECM protein genes including *Fn*, *Col1a1*, and *Col3a1* in control mouse lungs (21 days post-BLM treatment), which was significantly attenuated in *Txndc5^cKO* mice (Supplementary Fig. 12b). Taken together, these results demonstrate that inducing fibroblast-specific deletion of *Txndc5* significantly lessens the development and progression of BLM-induced PF and preserves lung function, suggesting that targeting TXNDC5 in vivo could be a powerful and effective therapeutic approach to mitigate lung fibrosis.

## Discussion
In this study, we demonstrated that TXNDC5, a lung fibroblast-enriched ER protein, plays a crucial role in the pathogenesis of PF. TXNDC5 is upregulated in the lung tissues/lung fibroblasts from both IPF patients and fibrotic mouse lungs induced by BLM treatment. Mechanistic investigations demonstrate that TXNDC5, a protein disulfide isomerase, promotes PF by augmenting TGFβ1 signaling activity through facilitating the folding and stabilization of TGFBR1 in lung fibroblasts, thereby resulting in excessive fibroblast activation, proliferation, and ECM production. In addition, TGFβ1-induced TXNDC5 upregulation in lung fibroblasts depends on increased ER stress and ATF6-mediated transcriptional regulation. Consistent with the in vitro mechanistic experiments, global deletion of *Txndc5* ameliorated BLM-induced lung fibrosis and respiratory dysfunction in mice. Importantly, induced fibroblast-specific deletion of *Txndc5* in established, BLM-treated fibrotic mouse lungs mitigated the progression of PF and preserved lung functions. Taken together, these results revealed a critical yet previously unrecognized role of TXNDC5 in the regulation of TGFβ signaling and in the pathogenesis of lung fibrosis. These data also suggest the therapeutic potential of TXNDC5 targeting in patients with PF. A schematic illustration of the regulation and function of TXNDC5 in modulating lung fibrosis is shown in Fig. 10h.

TGFβ1 is one the most potent and well-studied inducers of PF[20]. TGFβ1 signals are transduced by transmembrane serine/threonine kinase type II (TGFBR2) and type I (TGFBR1) receptors. TGFβ1 dimer first bind to TGFBR2, then TGFBR1 is recruited into the complex and phosphorylated/activated by TGFBR2. The intracellular signaling pathway of TGFβ receptors is mediated by Smad proteins (canonical pathway), as well as by other signaling molecules including MAPKs, JNK, p38, PI3K, and Rho family members (non-canonical pathways)[21,22]. Both canonical and non-canonical TGFβ signaling pathways contribute to the profibrogenic effects of TGFβ1, including fibroblast proliferation, fibroblast-to-myofibroblast differentiation, and ECM production/deposition[22]. Although TGFβ pathway is an attractive target for lung fibrosis, few candidates targeting TGFβ1 or TGFβ receptors directly have reached even early phase clinical trials[23]. Because TGFβ signaling plays an essential role in normal development, cell differentiation, and tissue homeostasis, broad inhibition of TGFβ signaling could lead to undesirable side effects, including liver[23] and cardiac toxicity[24,25]. In addition,

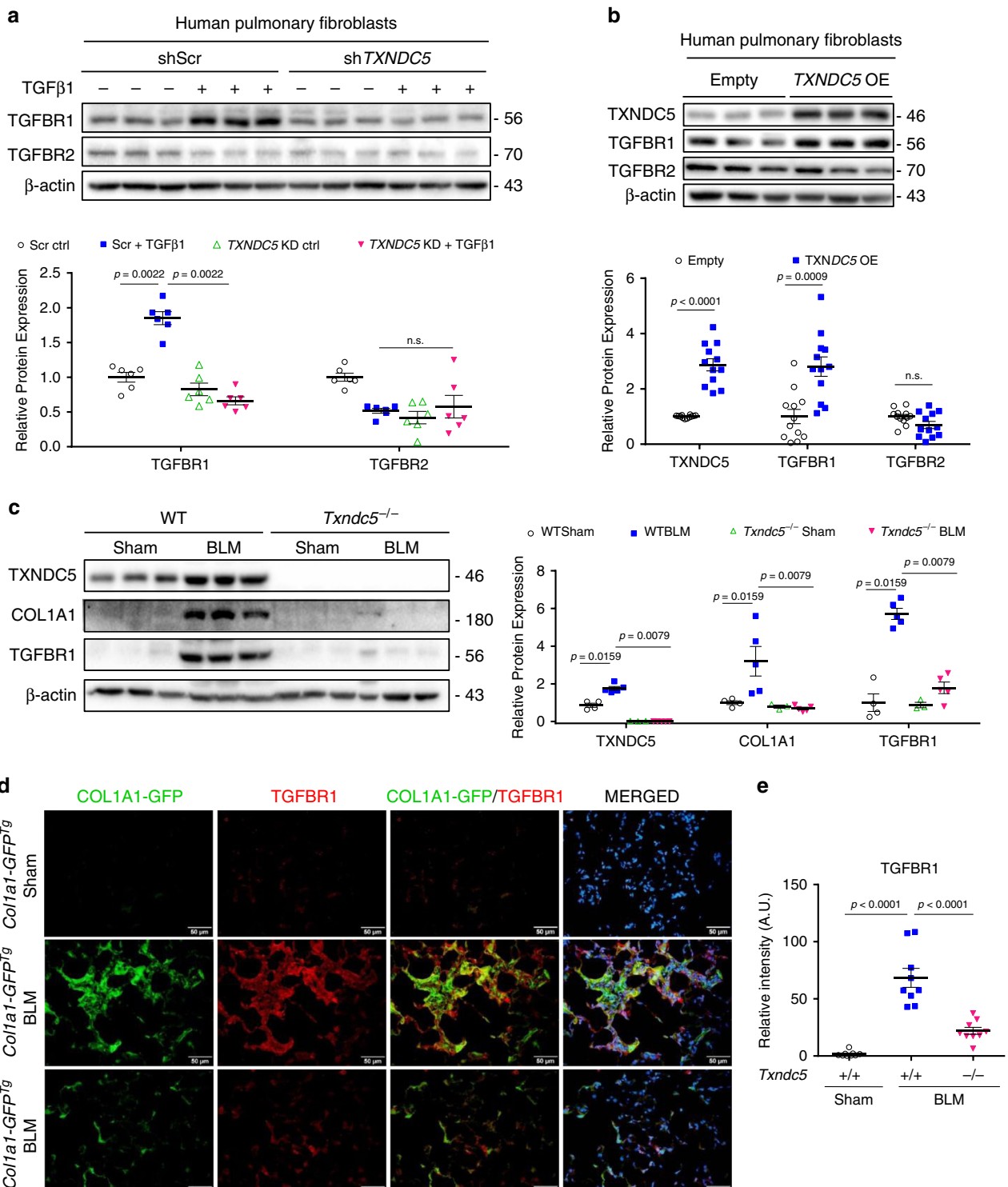

**Fig. 6 TXNDC5 modulates TGFBR1 expression in human lung fibroblasts and mouse lungs. a** Immunoblots showed that TGFBR1, but not TGFBR2, was markedly upregulated in control HPF (shScr) following TGFβ1 treatment. *TXNDC5* knockdown (sh*TXNDC5*) prevented TGFBR1 upregulation induced by TGFβ1 treatment completely ($n = 6$ biologically independent samples per group). **b** Forced *TXNDC5* expression led to marked upregulation of TGFBR1, but not TGFBR2, protein in HPF ($n = 12$ biologically independent samples per group). **c** TGFBR1 and COL1A1 proteins were both markedly upregulated in the lung tissues from WT, but not *Txndc5−/−*, mice 21 days following BLM treatment (WT sham $n = 4$, WT BLM $n = 5$, *Txndc5−/−* sham $n = 3$, *Txndc5−/−* BLM $n = 5$ biologically independent animals). **d** Representative IF staining and quantification (**e**) of TGFBR1 in *Col1a1-GFP^Tg* and *Col1a1-GFP^Tg\*Txndc5−/−* mouse lungs with PBS (Sham) or BLM treatment on day 21 ($n = 9$ fields examined over 3 biologically independent animals per group). TGFBR1 was marked increased and showed strong co-localization with GFP-positive lung fibroblasts in BLM-treated mouse lungs. Global deletion of *TXNDC5* prevented the upregulation of TGFBR1 in mouse lungs following BLM treatment (Data are presented as mean ± SEM, *P* value determined using two-tailed Mann–Whitney *U* test. Source data are provided as a Source Data file. n.s. non-significant, KD knockdown, OE overexpress, BLM bleomycin, TGFBR1 TGFβ receptor type 1, TGFBR2 TGFβ receptor type 2).

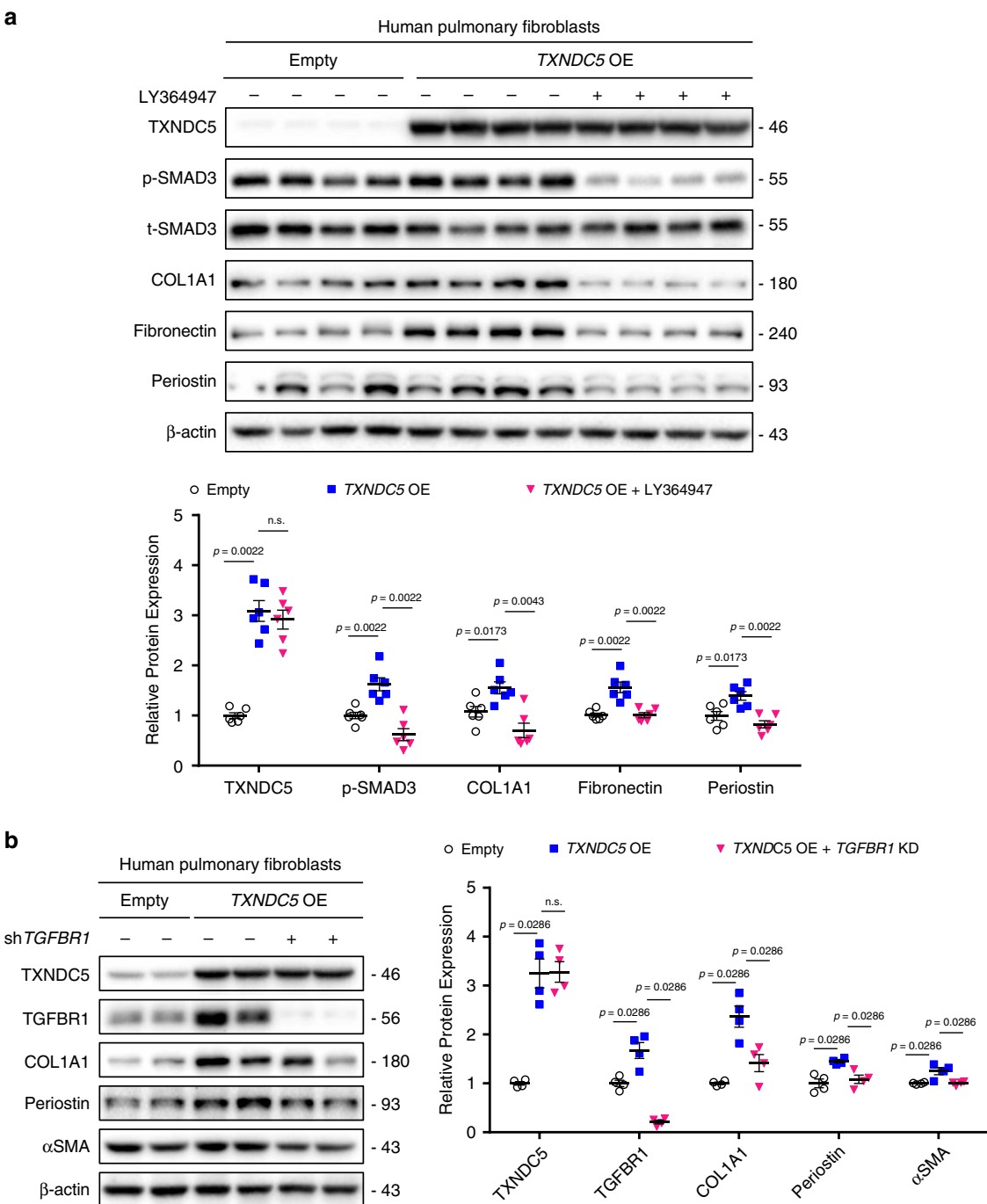

**Fig. 7 TXNDC5-induced pulmonary fibroblast activation and ECM production requires TGFBR1. a** Forced expression of *TXNDC5* in HPF led to increased SMAD3 phosphorylation, fibroblast activation (as evidenced by increased periostin levels) and ECM (COL1A1 and fibronectin) production, all of which were abolished completely by the treatment of LY364947 (10 μM for 48 h), a TGFBR1 kinase inhibitor (n = 6 biologically independent samples per group). **b** Knockdown of *TGFBR1* reversed *TXNDC5* overexpression-induced COL1A1, periostin, and αSMA expression (n = 4 biologically independent samples per group) (Data are presented as mean ± SEM, *P* value determined using two-tailed Mann–Whitney *U* test. Source data are provided as a Source Data file. n.s. non-significant, KD knockdown, OE overexpress, shTGFBR1 *TGFBR1* knockdown with shRNA).

*Tgfb1*-knockout mice showed increased systemic inflammation, perivasculitis, and interstitial pneumonia in the lungs[26], whereas *Tgfbr1*-deficient mice were embryonically lethal[27]. Conditional deletion of *Tgfbr1* in lung epithelium or mesenchyme also resulted in abnormal lung development[26]. Targeting TGFβ or TGFβ receptors directly, therefore, may not be a feasible therapeutic option for PF. The results presented here revealed a novel positive feedback loop of TGFβ1-ATF6-TXNDC5 signaling axis in lung fibroblasts, where TGFβ1 induces the upregulation of

TXNDC5 through increased ER stress and ATF6-mediated transcriptional control. Elevated TXNDC5 levels, on the other hand, further amplify TGFβ signaling by enhancing the folding and stability of TGFBR1. Targeting TXNDC5, therefore, could attenuate TGFβ signaling activity by breaking this positive feedback loop. In addition, modulating TGFβ signaling by TXNDC5 targeting avoids the risk of disturbing TGFβ-dependent physiological/cellular functions in non-fibroblast cells owing to its fibroblast-restricted expression pattern. In fact, deletion of

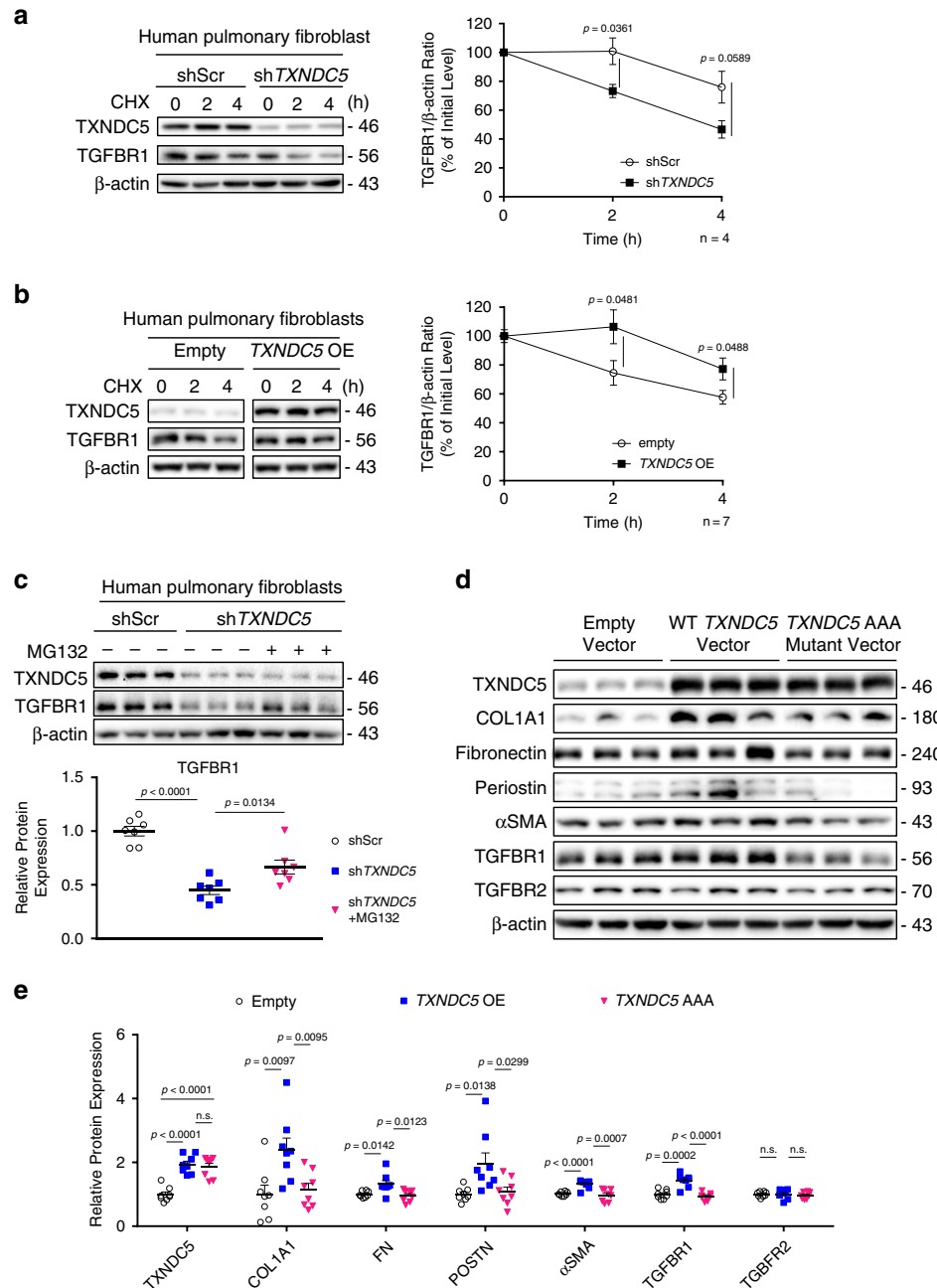

**Fig. 8 TXNDC5 promotes fibrogenesis by enhancing TGFBR1 protein stability via its PDI activity. a** A cycloheximide chase assay performed in HPF with *TXNDC5* knockdown (sh*TXNDC5*) showed accelerated degradation of TGFBR1 protein, comparing to control (shScr) ($n = 4$ biologically independent samples per group). **b** Overexpression of *TXNDC5* slowed down TGFBR1 protein degradation significantly in HFP ($n = 7$ biologically independent samples per group). **c** *TXNDC5* depletion-induced downregulation of TGFBR1 protein was partially reversed by the treatment of proteasome inhibitor MG132 (20μM for 48 h) ($n = 7$ biologically independent samples per group). **d, e** Overexpression of WT, but not AAA mutant (see text for details), TXNDC5 protein in HPF led to significant upregulation of TGFBR1, ECM (fibronectin and COL1A1) proteins and fibroblast activation markers (αSMA and periostin) ($n = 8$ biologically independent samples per group) (Data are presented as mean ± SEM, *P* value determined using two-tailed unpaired *t* tests. Source data are provided as a Source Data file. OE overexpress, TXNDC5 AAA TXNDC5 mutant lacking its PDI enzyme activity).

TXNDC5, either globally or specifically in fibroblasts, did not lead to discernible adverse effects on cardiac and pulmonary development, structure or function[14].

TXNDC5 is a member of the PDI family, an enzyme group that catalyzes the reduction and isomerization of disulfide bonds and facilitates the folding of nascent polypeptides[28,29]. The results presented here showed that TXNDC5-mediated TGFBR1 regulation and profibrogenic effects were dependent on its PDI activity (Fig. 8). This was the first report to show that TGFβ signaling

activity can be regulated by a PDI through modulating the folding and stability of TGFBR1, adding a new layer of molecular control of TGFβ signaling. Interestingly, re-analysis of the microarray data from IPF human lungs (GSE72073) and lung fibroblasts (GSE40839)[15] showed that among the 21 members of human PDI, only *TXNDC5* and *CASQ2* were significantly upregulated in human IPF, compared to control, lung tissue (Supplemental Fig. 13a). TXNDC5, however, was the only PDI that showed marked upregulation in the fibroblasts isolated from human IPF lungs

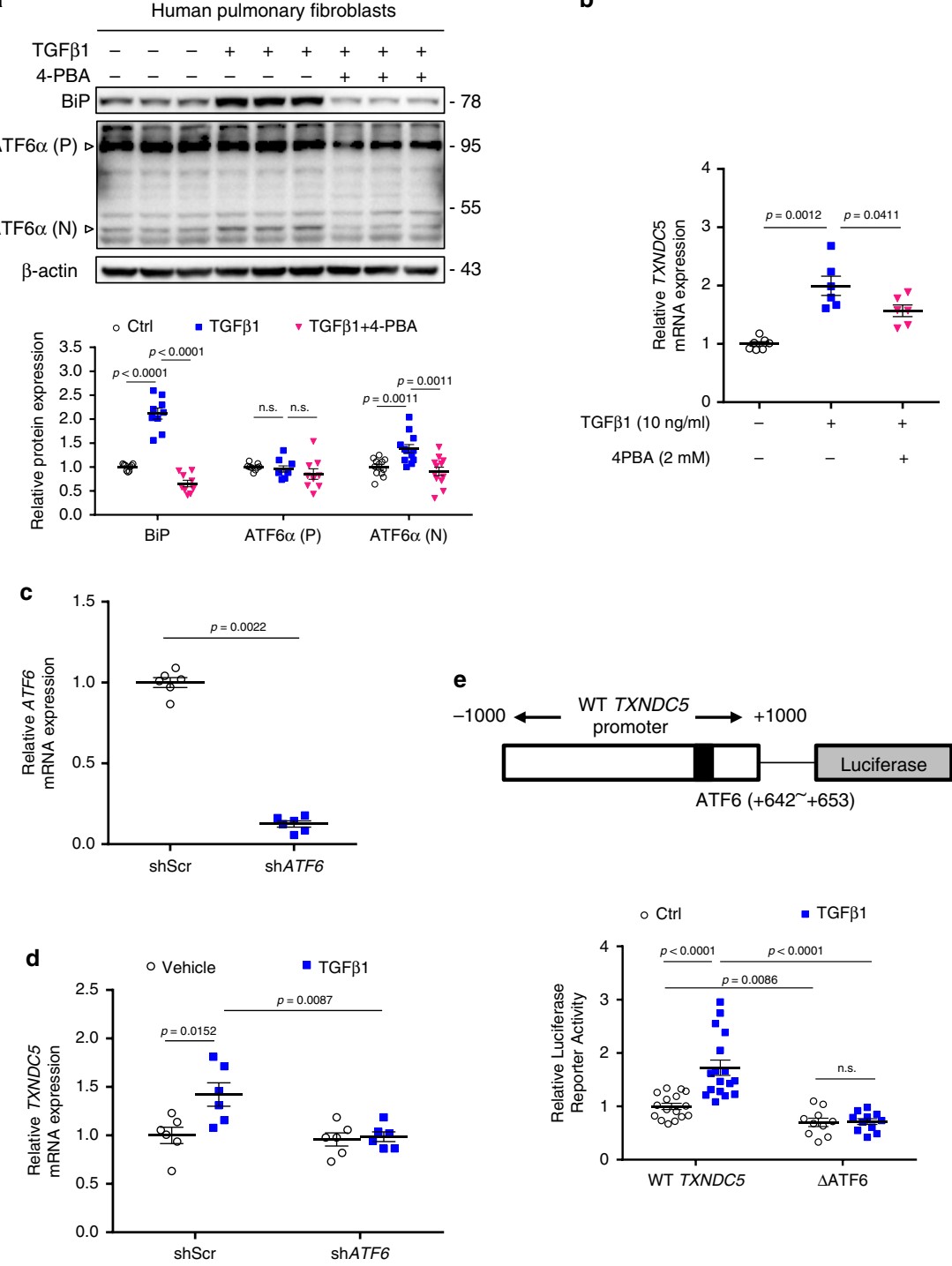

**Fig. 9 TGFβ1 induces TXNDC5 expression in HPF through ER stress-dependent ATF6 activation. a** TGFβ1 treatment in HPF led to increased ER stress levels, as reflected in upregulated ER stress markers BiP and ATF6α(N) (activated ATF6). Co-treatment with 4-PBA (2 mM), an ER stress inhibitor, reversed TGFβ1-induced increases in BiP and ATF6α(N) (Bip, ATF6α(P): $n = 9$ biologically independent samples per group, ATF6α(N): $n = 12$ biologically independent samples per group). **b** TXNDC5 mRNA was increased in response to TGFβ1 treatment, which was significantly attenuated by 4-PBA ($n = 7,6,6$ biologically independent samples). **c** Knockdown efficiency of lentiviral vectors carrying ATF6-targeted shRNA in HPF ($n = 6$ biologically independent samples per group). **d** TXNDC5 transcript was significantly increased in control (shScr), but not in ATF6-knockdown (shATF6), HPF following TGFβ1 treatment ($n = 6$ biologically independent samples per group). **e** Schematic illustration of the human TXNDC5 promoter luciferase reporter construct, which contains an ATF6-binding site. Deletion of the ATF6-binding site (TGACGTGG, $+642$ to $+653$, ΔATF6) markedly reduced the transcriptional activity of the TXNDC5 promoter in response to TGFβ1 stimulation (WT TXNDC5: $n = 16$ in ctrl, 17 in TGFβ1 biologically independent samples, ΔATF6: $n = 10$ in ctrl, 11 in TGFβ1 biologically independent samples) (Data are presented as mean ± SEM, P value determined using two-tailed Mann–Whitney U test. Source data are provided as a Source Data file. n.s. non-significant, ctrl control, shATF6: ATF6 knockdown with shRNA).

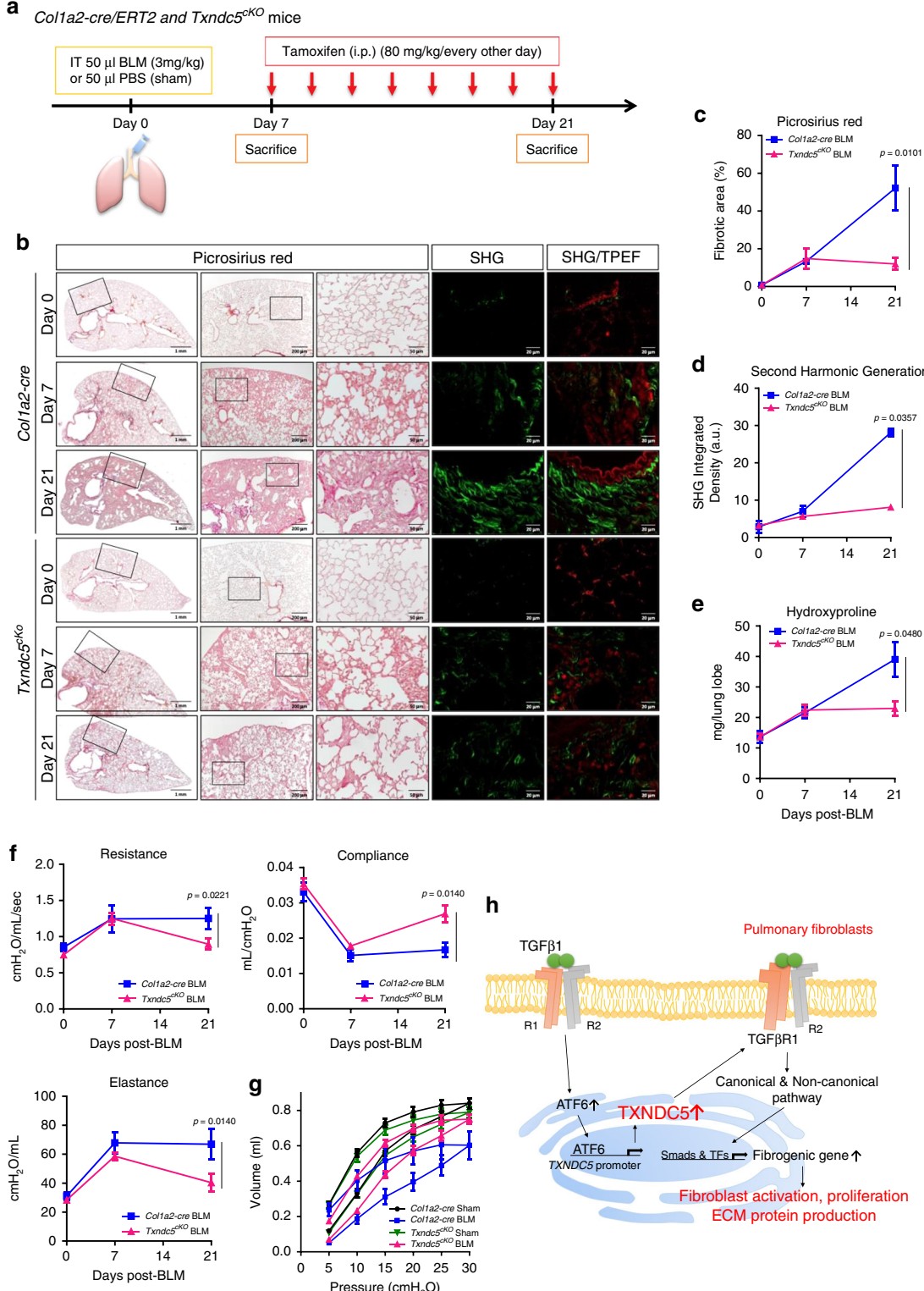

(Supplementary Fig. 13b), suggesting its fibroblast-specific function that distinguishes TXNDC5 from other human PDIs. Global deletion of TXNDC5 did not result in developmental defects or pathological changes, which could be explained by the functional redundancy among the PDI family proteins and the observed cell type-restricted expression pattern of TXNDC5. Consistent with this suggestion, *Txndc5* deletion did not lead to increased ER stress (Supplementary Fig. 14a–d) or apoptosis (Supplementary Figs. 14e, 16) in mouse lung tissue and fibroblasts.

The role of inflammation in IPF has been controversial. In most of the animal models of PF, it has been consistently observed that inflammation precedes fibrosis[30]. In addition, PF is often observed in the lung biopsy samples from patients with usual interstitial pneumonia resulting from chronic hypersensitivity pneumonitis, suggesting an inflammatory process preceding the development of fibrosis[31,32]. The extent of pulmonary inflammation, however, is generally minimal in IPF patients[31,32], and clinical trials using anti-inflammatory agents failed to

**Fig. 10 Fibroblasts-specific *Txndc5* deletion lessened the progression of pulmonary fibrosis. a** Illustration of experimental design for deletion of *Txndc5* in pulmonary fibroblasts. Tamoxifen (80 mg/kg i.p. every other day) was administered between 7–21 days after BLM treatment. **b** Picrosirius red staining (left panels) and second harmonic generation (SHG) images (right panels) of lung sections from *Col1a2-cre/ERT2* (*Col1a2-cre*) and *Col1a2-Cre/ERT2\*Txndc5^fl/fl^* (*Txndc5^cKO^*) mice 7 day and 21 day after intra-tracheal administration of BLM. The quantitative results of picrosirius red-(*Col1a2-cre*, $n = 3$ in D0 and D7, $n = 5$ in D21; *Txndc5^cKO^*, $n = 3$ in D0 and D7, $n = 6$ in D21 biologically independent animals) (**c**) and SHG- (*Col1a2-cre*, $n = 3$; *Txndc5^cKO^*, $n = 3$ in D0 and D7, $n = 5$ in D21 biologically independent animals) (**d**) positive areas showed rapid progression of PF in *Col1a2-cre*, but not in *Txndc5^cKO^*, lungs after BLM instillation. For each of the lung sections scanned for SHG, additional two-photon-excited fluorescence (TPEF) imaging was obtained to show the outline of the imaged tissue. **e** Hydroxyproline content was similarly increased in *Col1a2-cre* and *Txndc5^cKO^* on Day 7 (before Tamoxifen injection) following BLM treatment. Hydroxyproline content continued to increase in the mouse lungs of *Col1a2-cre* mice between 7 and 21 days post-BLM treatment (*Col1a2-cre*, $n = 3$ in D0, $n = 4$ in D7, $n = 5$ in D21; *Txndc5^cKO^*, $n = 3$ in D0 and D7, $n = 7$ in D21 biologically independent animals). The hydroxyproline content remained stable in BLM-treated *Txndc5^cKO^* mouse lungs after tamoxifen-induced *Txndc5* deletion. Representative lung function parameters (*Col1a2-cre*, $n = 5$ in D0, $n = 3$ in D7, $n = 6$ in D21; *Txndc5^cKO^*, $n = 4$ in D0, $n = 3$ in D7, $n = 7$ in D21 biologically independent animals) (**f**) and pressure-volume curves (on Day 21, sham: $n = 3$, BLM: $n = 6$ biologically independent animals) (**g**) determined using flexiVent FX system in *Col1a2-cre* and *Txndc5^cKO^* mice following sham procedure or BLM treatment. **h** Schematic summary of the proposed profibrotic mechanisms by which TXNDC5 contributes to the pathogenesis of pulmonary fibrosis (Data are presented as mean ± SEM, *P* value determined using two-tailed Mann–Whitney *U* test. Source data are provided as a Source Data file. BLM bleomycin). Illustrations in **a** and **h** were created by T.H.L.

improve outcome of IPF patients[33], raising the question about the contribution of inflammation to the pathogenesis of IPF. Our data showed that TXNDC5 was not detected in inflammatory cells in injured mouse lungs following BLM treatment. In addition, global deletion of *Txndc5* attenuated BLM-induced PF without affecting the extent of pulmonary inflammation (Fig. 4), suggesting little, if any, role of TXNDC5 in modulating inflammatory response in the lung tissue.

Using a global *Txndc5* knockout mouse line, we have shown that global deletion of *Txndc5* could "prevent" the development of lung fibrosis induced by BLM. In clinical settings, however, it is more important to develop a therapeutic approach that can halt or even resolve established PF in these patients. To this end, we generated a tamoxifen-inducible, fibroblast-specific *Txndc5* conditional knockout mouse line (*Txndc5^cKO^*). In these experiments, *Txndc5* deletion in lung fibroblasts was induced 7 days after BLM instillation, a time point when lung fibrosis was induced and began to expand, usually peaking at 2–3 weeks following BLM treatment in WT mice[34]. The results presented here demonstrate that depletion of *Txndc5* in lung fibroblasts significantly lessened the progression of lung fibrosis induced by BLM and prevented lung function from deteriorate. These data strongly suggest the potential of TXNDC5 targeting as a novel therapeutic approach to treat lung fibrosis.

Because TXNDC5, also known as EndoPDI, was initially characterized as a PDI enriched in the endothelium[29], it is not surprising that IF staining showed that TXNDC5 was also expressed in the endothelial cells of BLM-treated mouse lungs (Supplementary Fig. 2c, d). As endothelial cells have been suggested to contribute to the development of pulmonary fibrosis via endothelial-mesenchymal transition (EndoMT)[35,36], we could not exclude the possibility that TXNDC5 could also contribute to pulmonary fibrogenesis through promoting EndoMT in pulmonary endothelial cells. Further studies are required to test this hypothesis directly. The tamoxifen-inducible, fibroblast-specific conditional knockout mouse line (*Col1a2-Cre/ERT2\*Txndc5^fl/fl^*) used in the present study allows efficient deletion of *Txndc5* in collagen-producing, active lung fibroblasts regardless of their origins (i.e., from resident lung fibroblasts, endothelial cells or bone marrow-derived fibrocytes). The notion that fibroblast-specific deletion of *Txndc5* significantly lessened the development and progression of pulmonary fibrosis and lung dysfunction induced by BLM in mice, therefore, remains unchanged whether TXNDC5 contributes to EndoMT-mediated lung fibrogenesis or not.

Herein, we revealed a critical yet previously undiscovered role of the lung fibroblast-enriched ER protein TXNDC5 in the pathogenesis of lung fibrosis. The results presented here showed that TXNDC5 promotes lung fibrosis by enhancing TGFβ signaling activity via post-translational stabilization of TGFBR1 in lung fibroblasts, thereby leading to excessive lung fibroblast activation, proliferation, and ECM production. Targeted deletion of *Txndc5* protects against the development and progression of BLM-induced lung fibrosis. These results suggest that targeting TXNDC5 could be a powerful novel approach to ameliorate pulmonary fibrosis and respiratory dysfunction.

## Methods

**Ethical statement.** All studies were conducted in accordance with protocols approved by the institutional review boards of National Taiwan University and University of Chicago. All experimental animals were assigned unique identifiers to blind experimenter to genotypes and treatment. A block randomization method was used to assign experimental animals to groups on a rolling basis to achieve adequate sample number for each experimental condition.

**Human lung tissues.** All studies involving human lung tissues were approved by the University of Chicago IRB. Non-IPF control human lung samples were obtained from de-identified human lungs declined for transplantation (whose lungs were declined for transplantation, but whose next of kin had given consent for use in research) through the Regional Organ Bank of Illinois (ROBI) and Gift of Hope in collaboration with Dr. Julian Solway and Dr. Ann Sperling at the University of Chicago. IPF lung samples were obtained from de-identified explanted lungs of IPF patients undergoing lung transplantation at the University of Chicago. A written informed consent was obtained for each of the IPF patients. Multiple 0.5–1 cm biopsies from non-IPF and IPF lungs were dissected and flash-frozen in liquid nitrogen and stored at −80 °C until isolation of total RNA or protein. Grossly visible airways and blood vessels were avoided and dissected out of specimens when present.

Protein lysates were generated from lung tissues through homogenization of 50 mg tissue in 1 mL of deionized urea lysis buffer (8 M deionized urea, 1% SDS, 10% glycerol, 60 mM Tris pH 6.8, 0.002% pyronin Y, 5% β-mercaptoethanol) using a rotor-stator homogenizer. RNA was isolated from lung tissues by homogenization of 100 mg tissue in 1 mL TRIzol reagent (ThermoFisher) and purification of total RNA using the RNeasy Plus Mini Kit (Qiagen) according to the manufacturer's protocol.

***Txndc5^−/−^, Txndc5^fl/fl^,* and cell type-specific *Txndc5^cKO^* mice.** All experimental animals were maintained in a pathogen-free environment with a humidity of 40–70%, a temperature of 20–24 °C, and a 12-h light/dark cycle in the Laboratory Animal Center of National Taiwan University College of Medicine. *Txndc5^−/−^* mice were generated previously[14]. *Txndc5^fl/fl^* mice were generated using CRISPR/Cas9 genome editing. Cas9 mRNA, two single guide RNAs (sgRNA) targeting intron 1 and intron 3 of *Txndc5*, and two single strand donor oligodeoxynucleotides (ssONDs) carrying a loxP site to be knocked-in at introns 1 and 3, respectively, were co-injected into C57Bl/6 J mouse zygotes to generate mice with a floxed *Txndc5* (at Exon 2 & 3) allele via homology directed repair (Supplementary Fig. 8a). In short, the pX458 vectors expressing Cas9 and sgRNA targeting *Txndc5* genomic sequence were generated. The two sgRNAs flanking exon2 (sgRNA1) and exon3 (sgRNA2) of *Txndc5* were designed using CRISPR tool website (http://tools.genome-engineering.org). The resulting sequences are sgRNA1: 5′

ggaaacagaAATATCACACGTTTACTCGGaggtcaa3′ and sgRNA2: 5′tca-gaggttCAATCCA GTATCATCAAGGCaggaacatg 3′. T7 promoter sequence was then added to the Cas9 coding region and to the sgRNAs by PCR amplification. T7-Cas9 and T7-sgRNA PCR products were used as templates for in vitro transcription with mMESSAGE mMACHINE T7 ULTRA kit (Thermo Fisher Scientific, MA, USA). Both the Cas9 mRNA and the sgRNAs were purified by MEGAclear kit (Thermo Fisher Scientific, MA, USA). Two ssODNs carrying a loxP site that were to be inserted into intron 1 or intron 3 of *Txndc5* were designed: 5′ loxP ODN (for intron 1): 5′-TGAGCTCCAGGGGCACAACAAGCTATACGTTC CAGGAAACAGAAATATCACACGTTTACT**GAATTCATAACTTCGTATAAT GTATGCTATACGAAGTTAT**CGGAGGTCAATTAAAAGTCTAGGAGGCAG GGTCTGGCATAGAGGTCAAAAAAGAATACTA-3′, and 3′loxP ODN (for intron 3): 5′-GAGGATAACATTTAATTGGGGCTGGTGTATAGGTTCAGAG GTTCAATCCAGTATCATCAAx**GAATTCATAACTTCGTATAATGTATGCT ATACGAAGTTAT**GGCAGGAACATGGCAGCATCCAGGCAGGCATGGTGC AGAAGGAGCTGAGAGTTCTGTATC-3′. Purified Cas9 mRNA, sgRNA1, sgRNA2, 5′loxP OND and 3′loxP ODN were co-injected into one-cell mouse zygotes (C57BL/6/J) in M2 media (Millipore Corp. MA, USA) using a Piezo impact-driven micromanipulator. Post-injected blastocysts were transferred into the uterus of pseudopregnant female mice at 2.5 dpc. The preparation of mouse zygotes, pronuclei microinjection of Cas9 mRNA/sgRNAs, blastocysts transfer, and initial breeding of the *Txndc5*fl/fl animals were performed by the Transgenic Mouse Core Laboratory in National Taiwan University. PCR primers were designed to identify founders harboring two loxP knock-in sequences at the intended target site, as indicated by the presence of mutant PCR amplicons (5′loxP: 487 base pair, 5VF1 + 5VR1; 3′loxP: 449 base pair, 3VF2 + 3VR2, Supplementary Fig. 8b). The potential off-target mutageneses of CRISPR were assayed by RFLP/sequencing analysis at off-target sites predicted by CRISPR Design Tool. PCR, TA cloning, and DNA sequencing confirmed the presence of an allele with successful knock-in of two loxP sites that flanked exons 2 & 3 of Txndc5 in one of the founders. This founder was crossed to WT C57Bl/6 J mice to obtain *Txndc5*fl/+ offspring; the F1 *Txndc5*fl/+progeny were then crossed to generate homozygous *Txndc5*fl/fl mice. F2 *Txndc5*fl/fl mice were born at the expected Mendelian frequency and, at baseline, showed no detectable developmental defects or structural anomalies.

To generate an inducible, fibroblast-specific, conditional *Txndc5* mouse line, *Txndc5*fl/fl mice were bred with Col1a2-Cre/ERT2 transgenic mice (purchased from Jackson Laboratory, #029567). For activation of the Cre-ERT system, tamoxifen (80 mg/kg, dissolved in olive oil) was injected intraperitoneally every other day since day 7 and until day 21 after BLM treatment, both in control and *Txndc5*cKO mice.

**Reporter mouse models.** The endothelial cell (*Tie2-tdTomato*) and type II_pneumocyte (*SPC-mTmG*) -specific reporter mouse models were generated by crossing *Tie2-Cre/ERT2* with *ROSA26-tdTomato* mice and by crossing *SPC-Cre/ERT2* with *ROSA26-mTmG* mice, respectively. In these animals, endothelial cells and type II pneumocytes were labeled with tdTomato and GFP, respectively, after tamoxifen induction (80 mg/kg/day i.p. for 4 days). *Col1a1-GFP*Tg mice were kindly provided by Dr. David Brenner.

**Bleomycin-induced lung fibrosis model.** 8-to-10-week-old male WT (C57BL/6) mice and *Txndc5*−/− mice (and cKO mice) received intratracheal instillation of bleomycin (bleocin; Nippon Kayaku) at the dose of 3 mg/kg diluted in PBS or PBS only (sham). To collect bronchoalveolar lavage fluid (BALF), mice were sacrificed on day 7 and 14. To study fibrosis, lung samples were collected on day 21 for further analysis.

**In vivo microCT scanning.** Mice were sent to laboratory animal center in National Taiwan University for in vivo microCT scanning. Image reconstruction was performed using GPU-Nrecon Software. Ring artifact and beam-hardening correction were performed by this software as well. Reconstructed cross-sections were re-orientated and ROI was further selected. The analysis was performed according to methods of Bruker micro-CT. Lung volume was isolated and further quantified using CTAn software (1.16.4.1). The data were analyzed by Industrial Technology Research Institute (ITRI, Taiwan).

**Ex vivo microCT scanning.** Lung samples from WT and *Txndc5*−/− mouse treated with bleomycin were incubated for 2 h in 70, 80 and 90% ethanol consecutively and then overnight in 100% ethanol, followed by 100% hexamethyldisilazane for a further 2 h. The fixed mouse lung samples were sent to Industrial Technology Research Institute (ITRI, Taiwan) for further analysis. Briefly, the fixed mouse lung samples were scanned using high-resolution μCT scanner (SkyScan 2211, Bruker, Belgium). The voltage was 40 kVp and current was 700 μA with 180 degree scan. Image reconstruction was performed using fastest reconstruction software, Instarecon. X-ray grayscale thresholding and automatic ROI definition to carry out segmentation analysis of the lung tissue, where the volume of fibrotic and non-fibrotic lung areas was quantified for each segment; data for each lung (~900 sections) were then compiled into a composite measurement of fibrotic and non-fibrotic lung volumes.

**Lung function tests.** Lung function was assessed using the flexiVent system (Scireq, Montreal, QC, Canada). Mice were tracheostomized and ventilated at a rate of 150 breaths/min, tidal volume of 10 ml/kg, and a positive end-expiratory pressure (PEEP) of 2–3 cmH₂O. A deep inflation perturbation was used to estimate the inspiratory capacity (IC). Pressure-Volume loops were generated by constant increasing pressure, followed by regular decreasing pressure. Other lung function parameters like resistance, compliance, and elastance were measured by using SnapShot-150.

**Bronchoalveolar lavage.** WT and *Txndc5*−/− mice were subjected to bronchoalveolar lavage thrice with 0.5 ml of 0.9% NaCl. The BALF samples were collected and centrifuged at 500 g for 5 min, and the supernatant of the lavage fluid was used to measure total protein concentration using BCA Protein Assay Kit (Thermo Fisher Scientific, MA, USA). Differential cell counts were performed using cytospin preparations, followed by Differential Quick Stain Kit (Modified Giemsa) staining (Polyscience, IL, USA). More than three hundred cells were counted in bleomycin-treated group and about one hundred cells were counted in sham group under 400-fold magnification, and the absolute number of each cell type were calculated.

**Hydroxyproline assay.** Levels of hydroxyproline in lung tissues were measured using the Hydroxyproline Colorimetric Assay kit (BioVision, Milpitas, CA, USA). Briefly, 10 mg of frozen right middle lobes homogenized in 100 μL of hydrochloric acid (HCl, 12 N), and hydrolyzed at 120 °C for 3 h. Then, 10 μL of individual samples were used to quantify the absorbance at 560 nm. Hydroxyproline content was presented in milligram per lobe.

**Histology.** Mouse left lungs were fixed overnight in 4% paraformaldehyde (PFA), embedded in paraffin, and sectioned in 5 μm thickness for picrosirius red (Abcam, Cambridge, UK) and Masson's trichrome staining (Sigma Aldrich, MO, USA). Measurement of fibrotic area was quantified using ImageJ software (NIH, http://rsbweb.nih.gov/ij/).

**Immunohistochemical staining.** Immunohistochemical (IHC) staining was carried out using Novolink™ Polymer Detection System (Leica Biosystems, Wetzlar, Germany) as described previously[14]. In short, mouse left lung tissues were embedded in paraffin, sectioned and deparaffinized followed by antigen retrieval. The endogenous peroxidase was neutralized using Peroxidase Block. Mouse lung sections were then incubated with primary antibodies including anti-αSMA (1:400, Abcam, Cambridge, UK, ab5694) and anti-TXNDC5 (1:1500, Proteintech, IL, USA, 19834-1-AP) overnight at 4 °C. Sections were then washed and incubated with Post Primary for 1 h at room temperature, followed by incubation with Novolink™ Polymer for 15 min. Sections were developed with DAB working solution and then counterstained with hematoxylin and mounted with mounting medium. Staining area quantification was performed using ImageJ.

**Immunofluorescence.** Mice were perfused intracardially with PBS at room temperature. Mice lungs (right inferior lobes) from PBS- and bleomycin-treated WT mice and *Col1a1-GFP*Tg mice (a gift from Dr. David Brenner at UCSD)[16] were fixed with 4% PFA for 4 h, sectioned in 5 μm thickness, permeabilized and blocked with 0.1% Tween 20 in 2% bovine serum albumin (BSA) for 1 h. Fixed lung sections were then incubated with the primary antibody, rabbit anti-TXNDC5 (1:1500, Proteintech, IL, USA, 19834-1-AP), anti-TGFBR1 (1:100, OriGene, MD, USA, AP14647PU-N) and/or mouse anti-SMA (1:500, Sigma Aldrich, MO, USA, A5228) at 4 °C overnight. After washing, the sections were incubated with Alexa Fluor 594-labeled anti-rabbit secondary antibodies and/or Alexa Fluor 488-labeled anti-mouse (1:500, BioLegend, San Diego, CA) at room temperature for 1 h. The sections were then washed and mounted with ProLong Gold (Thermo Fisher Scientific, MA, USA). The fluorescence image was acquired with fluorescence microscope Olympus BX51 combined with Olympus DP72 camera and cellSens Standard 1.14 software (Olympus, Germany) and analyzed using ImageJ. The objective was Olympus UPLFLN 40X Objective without chromatic aberrations correction, and the filters were U-MWB2 (FITC) (Ex/Em:460-490/520IF (nm)), U-MWG2 (TRITC) (Ex/Em:510-550/590 (nm)), and U-MWU2 (DAPI) (Ex/Em:330-385/420 (nm)). The excitation source was a mercury lamp. The coverslips were from Deckglaser Microscope Cover Glass, and the thickness was 0.13–0.16 mm.

**Human pulmonary fibroblasts (HPF) culture.** Primary adult human pulmonary fibroblasts (ScienCell, CA, USA, 3310) were cultured in Fibroblast Medium (ScienCell, CA, USA, 2301) containing 2% FBS, 1% Fibroblast Growth Supplement (FGS) and 1%penicillin/streptomycin solution (P/S) in a humidified atmosphere of room air supplemented with 95% O₂/5% CO₂ at 37 °C.

**NIH-3T3 mouse fibroblasts culture.** NIH-3T3 mouse fibroblasts were purchased from the American Type Culture Collection (ATCC) and cultured in DMEM with the addition of 10% FBS, 100 IU/ml penicillin G/100 μg/ml streptomycin/0.25 μg/ml amphotericin and plated in a culture flask maintained in an incubator at 37 °C in a humidified atmosphere of room air supplemented with 95% O₂/5% CO₂.

**Murine primary lung fibroblasts (MLF) isolation**. Four-week-old male WT (C57BL/6 J) and $Txndc5^{-/-}$ mice were anaesthetized and the lungs were harvested immediately using aseptic procedures. Lungs were washed clean in PBS (phosphate buffer saline) and digested with 0.14 Wunsch units/ml Liberase (Sigma Aldrich, MO, USA) at 37 °C for 1.5 h with shaking. The reaction was stopped with MEM (Thermo Fisher Scientific, MA, USA) containing 10% FBS. The samples were centrifuged at 520 g for 5 min and then resuspended with MEM thrice to remove the traces of Liberase. Pellets were resuspended in MEM containing 10% FBS, 1X penicillin-streptomycin, sodium bicarbonate, and sodium pyruvate. Freshly isolated and up to second passage of mouse lung fibroblasts were used in experiments.

**TGFβ1 stimulation in fibroblasts**. HPF and MLF were cultured in serum free medium and treated with TGFβ1 (PeproTech, NJ, USA) at 10 ng/ml for 48 h, then processed for downstream experiments. For assaying SMAD3 activation in response to TGFβ1 stimulation, cell lysates were collected 6 h after TGFβ1 treatment.

**Viral transduction for gene knockdown and overexpression**. Lentiviral particles containing shRNAs pLKO-sh*TXNDC5* (#TRCN000033258), shRNAs pLKO-sh*TGFBR1* (#TRCN0000196326), shRNAs pLKO-sh*XBP1* (#TRCN0000019804), and shRNAs pLKO-sh*ATF6* (#TRCN0000416318) was used to knockdown *TXNDC5, TGFBR1, XBP1, ATF6* respectively in HPF. Lentivirus containing scrambled shRNA, pLKO-shScr (#TRCN00001), was used as non-targeting control. To overexpress *TXNDC5*, lentiviral particles containing human *TXNDC5* gene (pLAS2w.pPuro-TXNDC5) were used and an empty pLAS2w.pPuro vector were used as control. Transduced HPF were treated with Fibroblast Medium containing 1.5 ng/ ml puromycin for selection of transduced cells.

**siRNA for *EIF2A* knockdown in HPF**. *EIF2A* was knocked down in HPF using human *EIF2A*-targeting siRNA duplexes designed using the sequence from the open reading frame of human *EIF2A* (5′-AACCACAATCAGGAAACGATA-3′). Scrambled oligo-ribonucleotide complex (siScr) not homologous to any mammalian genes was used as control. Cells were transfected with Lipofectamine RNAiMAX (Thermo Fisher Scientific, MA, USA) according to manufacturer's instructions. Knockdown efficiency was assayed using qPCR.

**RNA extraction and qRT-PCR**. Total RNA was isolated from lung tissues and cells using TRIzol (Thermo Fisher Scientific, MA, USA) according to the manufacturer's recommendations. Total RNA was reverse transcribed with Maxima First Strand cDNA Synthesis Kit (Thermo Fisher Scientific, MA, USA) and SYBR Green qRT-PCR was performed[14,37]. The expression level of each individual transcript was normalized to control gene HPRT and expressed relative to the mean expression values of control samples. The primers are list in Supplementary Table 1.

**Immunoblot analysis**. Lung tissues and cells were homogenized using 1x Cell Lysis Buffer (Cell Signaling Technology, MA, USA) supplemented with protease inhibitor and phosphatase inhibitor cocktail (MedChemExpress, NJ, USA, HY-K0010P-1, HY-K0021, HY-K0022), centrifuged at 4 °C for 10 min at 10,000 × g, and the supernatant was mixed with 4X Protein Loading Buffer, followed by boiling at 95 °C, 5 min. Protein samples were fractionated on 8% SDS-PAGE gel, transferred onto PVDF membrane and then blocked in blocking buffer (5% BSA, 0.1% Tween 20 in PBS). Membranes were incubated with primary antibodies against COL1A1 (1:1000, EMD Millipore, CA, USA, AB765P, for mouse species), COL1A1 (1:1000, OriGene, MD, USA, TA309096, for human species), FN (1:1000, BD Biosciences, CA, USA, 610077), TXNDC5 (1:15000, Proteintech, USA, 19834-1-AP), ELN, POSTN XBP-1u, ATF4(1:1000, GeneTex, CA, USA, GTX37428, GTX100602, GTX113295, GTX101943), total/p-JNK, total/p-ERK, total/p-38, BiP, XBP-1s, CHOP (1:1000, Cell Signaling Technology, MA, USA, 9252, 9251, 9102, 4370, 9212, 9211, 3177, 12782, 2895), αSMA, total/p-SMAD3, HA-tag (1:1000, Abcam, Cambridge, UK, ab5694, ab52903, ab40854, ab9110), TGFBR1 (1:1000, Thermo Fisher Scientific, MA, USA, PA5-32631, for human species), TGFBR1 (1:1000, Abcam, Cambridge, UK, ab31013, for mouse species), TGFBR2 (1:1000, OriGene, MD, USA, TA311643), ATF6 (1:1000, Bio Academia, Osaka, Japan, 73–505), Flag-tag (1:1000, Cell Signaling Technology, MA,USA, 8146), β-actin (1:1000, OriGene, MD, USA, TA811000) overnight at 4 °C. Blots were developed using HRP-conjugated anti-mouse or anti-rabbit whole IgG secondary antibodies (1:10000, Thermo Fisher Scientific, MA, USA) and WesternBright ECL HRP substrate (Advansta, CA, USA). Protein band detection was performed using ChemiDoc MP system (BioRad Laboratories, CA, USA). Protein band intensity quantification analysis was performed with ImageLab software version 5.1. The full scans of all western blots with molecular markers are performed in Supplementary Fig. 15.

**Fibroblast proliferation assay**. Fibroblast proliferation activity was measured using Cell Proliferation ELISA, BrdU (colorimetric) (Sigma Aldrich, MO, USA). Fibroblasts (HPF and MLF) were labeled with 10 μmol/L bromo-deoxyuridine (BrdU) in a 96-well plate for 24 h at 37 °C. DNA synthesis was assessed as a surrogate for cell proliferation using a colorimetric ELISA according to the

manufacturer's instructions. Absorbance of the samples was measured in a spectrophotometer at 370 nm (reference wavelength 492 nm).

**Protein stability assay**. TGFBR1 protein stability was measured using cycloheximide pulse chase assay. Control and *TXNDC5*-knockdown or *TXNDC5*-overexpression HPF were treated with 100 μg/ml cycloheximide (Cayman Chemical, MI, USA, 14126) to inhibit protein translation. TGFBR1 protein levels were determined in different time points and expressed as percentages relative to time 0. β-actin was used as a control protein.

**TXNDC5 promoter luciferase activity assay**. TXNDC5 promoter luciferase activity assay was implemented according to the Luc-Pair™ Duo-Luciferase Assay Kit 2.0 user manual (GeneCopoeia, MD, USA). To investigate the activity of *TXNDC5* promotor with (pGL3-h*TXNDC5*, +642 to +653 deleted) or without (pGL3- h*TXNDC5*, −1000 to +1000) deletion of its ATF6 binding site, constructs were generated by the BioMed Resource Core of the 1st Core Facility Lab, NTU-CM. HPF cells were transfected with each plasmid for 24 h followed by 48 h TGFβ1 (10 ng/ml) treatment. Signal intensity was evaluated in a plate reader (Synergy HT, BioTek, VT, USA). In these experiments, a Renilla luciferase expression vector was co-transfected and used as a control for transfection efficiency.

**Generation of TXNDC5 AAA mutant lacking PDI activity**. To generate a mutant *TXNDC5* construct with cysteine-to-alanine mutations in both ends of each of its 3 thioredoxin domains (CGHC to AGHA), which are required for the PDI activity of TXNDC5, a full-length WT TXNDC5 cDNA clone (pLAS2w.Ppuro-hTXNDC5-AAA) was subjected to site-directed mutagenesis using the QuickChange II Site-Directed Mutagenesis kit (Agilent Genomics, CA, USA) according to the manufacturer's instructions. To overexpress *TXNDC5* AAA mutant, lentiviral particles containing pLAS2w.Ppuro-hTXNDC5-AAA were used, whereas lentiviral particles containing an empty pLAS2w.pPuro vector were used as control. Transduced HPF were treated with Fibroblast Medium containing 1.5 ng/ml puromycin for selection of transduced cells.

**In situ PLA**. To detect protein–protein interaction, Duolink® in situ PLA (Sigma Aldrich, MO, USA) was performed according to the manufacturer's recommendations. HPF grown on chamber slide were fixed in 4% PFA for 10 min, permeabilized and blocked with 0.1% Tween 20 in 2% bovine serum albumin (BSA) for 1 h, and then incubated with primary antibodies, rabbit anti-TXNDC5 (1:1500, Proteintech, IL, USA, 19834-1-AP) together with mouse anti-TGFBR1 (1:100, Abnova, Taipei, Taiwan, H00007046-A01) or mouse anti-TGFBR2 (1:100, Abcam, Cambridge, UK, ab78419) overnight at 4 °C. After washing, slides were incubated with PLA probe for 1 h at 37 °C, followed by ligation and amplification. The fluorescence image was captured by fluorescence microscope. Slides without primary antibodies were used as negative control.

**Co-immunoprecipitation (Co-IP)**. Human WT *TXNDC5*, HA-tagged mutant TXNDC5-AAA and human *TGFBR1-Myc-Flag*-tagged mRNA were transfected into NIH-3T3 fibroblasts using Lipofectamine MessengerMAX (Thermo Fisher Scientific, MA, USA) for 4 h according to manufacturer's instructions. Protein lysates from NIH-3T3 were incubated with mouse anti-Flag tag monoclonal antibody (Cell Signaling Technology, MA,USA, #8146) or mouse IgG isotype control (Cell Signaling Technology, MA,USA, #5415 S) with protein G magnetic beads and collected with a magnetic stand. IP of WT or mutant TXNDC5 was conducted using a magnetic IP kit (Thermo Fisher Scientific, MA, USA). Proteins co-immunoprecipitated with Flag-tag, TXNDC5, or HA were eluted and subjected to gel electrophoresis and immunoblotting using the antibodies described above.

**Apoptosis detection**. The extent of cell apoptosis in lung fibroblasts isolated from WT and *Txndc5* knockout mice was determined using BD Annexin V:FITC Apoptosis Detection kit (BD Biosciences, NJ, USA) on a BD FACSCalibur™ flow cytometer following the manufacturer's instructions. The data acquisition, analysis and reporting were performed using BD FACSuite software (BD Biosciences, NJ, USA).

**Multiphoton microscopy and SHG imaging**. The second harmonic generation (SHG) imaging of the lung tissue sections were obtained using a multi-photon microscopic system[38,39]. In short, a wavelength-tunable Ti:Sapphire laser (Mai-Tai® DeepSee Tsunami; Spectra Physics, Mountain View, CA) pumped by a diode-pumped solid-state laser (Millennia X; Spectra Physics, Mountain View, CA) was used as the excitation source. The 920 nm output of the Ti:sapphire laser is scanned in the focal plane by a galvanometer-driver x-y mirror scanning system (Southport Corp., Taiwan). For high resolution imaging, high-numerical-aperture water-immersion 20X/NA1.0 objective lens (XLUMPHLFLN-W, Olympus, Japan) was used. To optimize image quality without laser ablation, the laser power was set to be about 70 mW on the tissue section. In addition, the SHG signals were spectrally separated by bandpass filters of 470/22 nm (FF01-470/22, Semrock, USA). The resolution of scan format was 1024 × 1024 pixels. For each of the lung sections scanned for SHG, additional two-photon-excited fluorescence (TPEF) imaging was obtained to show the outline of the imaged tissue.

**Statistics and reproducibility**. All experimental data were reported as means ± SEM. The statistical significance of differences between experimental groups was evaluated by Mann–Whitney *U* test or unpaired *t* test. *P* values < 0.05 were considered statistically significant. All data (except for Supplementary Figs. 2c–g and 8c, d) shown were obtained from at least 3 biological independent experiments. Supplementary Figs. 2c–g and 8c, d were performed by 2 biological independent experiments.

**Reporting summary**. Further information on research design is available in the Nature Research Reporting Summary linked to this article.

## Data availability

The data that support the findings of this study are available from the corresponding author upon reasonable request. Microarray data of human lung tissues (GSE72073) and human lung fibroblasts (GSE40839) are from the public GEO database. Source data are provided with this paper.

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

## Acknowledgements

The authors would like to thank Dr. Julian Solway M.D., Kelly Blaine M.S., Katie O'Halloran R.N., and Cortney Bond R.N. for their assistance in obtaining human lung specimens used in this study. The authors are also indebted to the donors, families, and staff of the Gift of Hope Organ and Tissue Donor Network for their generous donation of lung tissues used in this study. We thank the staff of the Biomedical Resource Core at the First Core Labs, National Taiwan University College of Medicine, for technical assistance. We also thank the technical services provided by the Transgenic Mouse Model Core facility and the National RNAi Core Facility at Academia Sinica, Taiwan, This work was funded by a Taiwan Ministry of Science Technology Grants 103-2320-B-002-068-MY2 (KCY), 105-2628-B-002 -042-MY4, 108-2314-B-002-199-MY3 (KCY), 108-2314-B-002-155-MY3 (PNT), 106-2314-B-002-163-MY3 (CCW); Taiwan National Health Research Institute Career Development Grant NHRI-EX104-10418SC and Innovative Research Grant NHRI-EX109-10936SI (KCY); a CRC Translational Research Grant from the Institute of Biomedical Sciences at Academia Sinica, Taiwan IBMS-CRC108-P03 (KCY) and grants from National Taiwan University Hospital NTUH. 109-T15 (PNT), 106-P02, 105-CGN01, UN106-026, 106-N3740, VN106-12, 107-T02, UN107-019, 107-N4062, VN107-03, 108-T16, VN108-06, VN109-07, NTUH.108-P04, 108-N4198, 108-S4247, 108-EDN03, 109-S4576 (KCY) and a Career Development Grant from National Taiwan University 109L7872 (KCY). These funding agencies had no role in the study design, data collection/analyses, preparation of the manuscript or decision to publish.

## Author contributions

Conceptualization: T.H.L, Y.T.L, Y.F., and K.C.Y.; experimental design: T.H.L., Y.T.L., Y.F., C.F.Y., R.D.G., C.C.W., C.N.C., and K.C.Y.; data collection and analyses: T.H.L., Y.T.L., C.F.Y., Y.C.S., Y.T.C, C.T.H., F.L.L., P.N.T., T.H.L., M.Y.Y., P.C.W., T.P.S., R.T.H., Y.S.L., S.C.L., Y.F.W., S.J.L., Y.S.T., P.C.W., W.L.W., S.L.L., R.D.G., Y.F., and K.C.Y.; creation of illustrations in 10a and 10 h: T.H.L.; manuscript writing, review, and editing: all authors.

## Competing interests

The authors declare no competing interests.

**Additional information**

