## [Peer Review File · Nature Communications]

Reviewers' comments:

Reviewer #1 (Remarks to the Author):

1. Please make sure that the figure legends for animal studies point out clearly that (eg Figure 2) the tissue extracts and other histology are from day 21 post bleo animals.

2. The data in figure 2 show mRNA levels at day 21 of bleo treatment. How does this data compare to the several array data sets in the literature already?? (Kaminski et al 2000; Haston et al 2005 or Cabrera et al 2013).

3. Supplementary figures appear to be added to previous concerns about localization of TXNDC5 to fibroblasts only. Endothelial cells appear positive (Supplementary fig 2c and 2d) so what endothelial cell functions or productions are affected in the global KO mice that could affect fibrogenesis??

4. Figure 3d implicates TXNDC5 in fibroblast accumulation after bleo, but what is the effect of TXNDC5 deletion on other fibroblast functions in other tissues?? (tissue repair, normal growth, female post menstruation restructuring, etc).

5. Fig 3b what does "white areas" refer to? How is it detected? How does this show lung volume reduction?

6. Also for 6e the details refer to "Targeted deletion of Txndc5" when it is a global deletion and this same usage is in second to last paragraph of discussion "Using global TXNDC5 knockout mouse line we have shown that targeted deletion of TXNDC5 " can be confusing as to what is the "targeted" cell? In my understanding, targeted implies the gene is being deleted in a specific cell. The experiment to delete in lung fibroblasts (Fig 5 a and b) specifically uses fibroblast in culture and cannot be universally applied or construed to relate to fibroblast specific deletion in vivo.

Very confused aggregation of data in combining individual cell cultures alongside in vivo Bleomycin experiments.

7. Fig 6 TXNDC5 knockdown should be shown to be effective by decrease or absence of TXNDC5 mRNA. Again, the use of "targeted deletion" (fig 6d) is confusing and missing the day 21 detail that is needed to clarify the situation in the bleomycin experiment. It was a global deletion mouse that was used.

8. The claim is that TXNDC5 facilitates the folding of TGFbR1, but what is the effect of this chaperone on TGFbR2?? Supp figure 7 shows the FRET data for TGFbR1 but what is the data for TGFbR2?? Both are required for proper TGFb1 signalling??

9. Fig 7b: If TXNDC5 acts through TGFbR1 and not 2 there should be R2 dependent genes that are regulated through TGFb stimulation. You should show this to certify that R1 is the target even though it is recognized that R2 forms a heterodimeric complex with TGF-beta receptor type-1, and binds TGF-beta to phosphorylate proteins, which then enter the nucleus and regulate the transcription of genes related to cell proliferation, cell cycle arrest, wound healing, immunosuppression, and tumorigenesis. Are any or all TGFb responsive genes depleted by KO of R1?

Also co-precipitation is a usual way to show direct protein-protein interaction but Fret analysis seems good for conclusion.

10. For a Nature article in 2019, the background references regarding the myofibroblast and pulmonary fibrosis seem somewhat dated (ref 2,3 and 7). There has been much written about these 2 issues since 2003 and 2013.

11. The most crucial question regarding whether fibroblast specific deletion (therefore fibroblast targeted deletion) of TXNDC5 leads to inhibition of fibrogenesis is addressed in Fig 10 and suggests strongly that this is an important pathway in progressive fibrosis. Critical questions revolve around whether this pathway of intervention in TGFb1 mediated fibrosis is any different from those examined through the use of other ALK 5 inhibitors, (such as LY 364947 or SD201) and can avoid the cardiac toxicity (at least) seen with both of these very powerful TGFb1 inhibitors??

Reviewer #2 (Remarks to the Author):

The study by Lee et al seeks to understand the pathogenic mechanisms underlying the progression of pulmonary fibrosis. The study's major focus is on the protein TXNDC5, a disulfide isomerase abundantly expressed in the ER of mammalian cells. Interest in this protein relates to a recent report by this group showing that deficiency of TXNDC5 exacerbates cardiac fibrosis in rodents. In this study, the focus is on TXNDC5 expression in the lung. The major findings are that levels of TXNDC5

are increased in lungs of IPF patients as well as bleomycin-exposed mouse lungs, and that global or fibroblast-specific deficiency of TXNDC5 ameliorates experimentally induced pulmonary fibrosis in mice. Mechanistically, investigators also purport that TXNDC5 is critical for regulating myofibroblast differentiation by controlling the expression of TGFbeta receptor 1 levels. Overall, findings in this publication are likely to be of interest to the wide readership audience of this journal, although conclusions are somewhat oversimplified and seem to argue against the current paradigm of ER stress playing a pathogenic role in fibrotic lung diseases. To support the conclusions of the manuscript the following studies should be considered.

Major

- 1) The mechanistic connection between TXNDC5 and ER stress is rather weak. Please examine more than one ER stress pathway, namely PERK and IRE and their downstream effectors as well as consider verifying that protein aggregation is increased in cells.
- 2) As a control, please knockdown either the PERK or IRE pathways (regardless of whether elevated) to confirm that TXNDC5 is not regulated through other arms of the unfolded protein response. Otherwise, it is somewhat premature to conclude that everything is regulated through ATF6.
- 3) The idea of knocking down TXNDC5 to reduce pulmonary fibrosis is somewhat confusing since one would expect that ER stress would be increased in the absence of this proteins expression (cells would have a harder time folding proteins). To address this, please examine markers of ER stress in lung and fibroblasts of TXNDC5 expressing and non-expressing cells.
- 4) Figure 2 B and C do not permit adequate visualization of TXNDC5 expression in type II cells, which are also known to upregulate UPR machinery in response to bleomycin. Please provide low power images of lung under bleomycin conditions to confirm that TXNDC5 is not expressed in these cells.
- 5) Please confirm that TXNDC5 deficiency fibroblast are still viable. Otherwise, it is possible that increased cell death explains the reduction in myofibroblast markers after knockdown of this protein.

Reviewer #3 (Remarks to the Author):

The authors have performed a number of assays to report a connection between expression of a protein disulfide isomerase and pulmonary fibrosis. There is an interesting story here. However, there are some issues with internal consistency and the proposed mechanism. There's clearly an effect on lung tissue and cells. Whether it is related to TGFB1R is debatable.

My main concerns are to change the histograms to scatter plots to more accurately present the data, provide more details about some of the imaging assays, reconcile differences between experiments, and to get rid of the deeply flawed FRET assay. Details are listed below.

Fig. 1. Rather than cover up the noisiness of the data with block histograms, the authors should present scatter plots that show the range of the data. Please state whether errors are reported as SD or SEM.

Fig. 1b. I disagree with the conclusion that there is a strong correlation. Collagen levels are highly variable relative to TXNDC5 levels. The R value reflects an averaging effect, not a good fit.

Fig 1e. FN data inclusion is questionable. There does appear to be a difference between low and high TXNDC5 expression and FN levels, though the FN level differences are quite modest, maybe 30% at best. That seems too low to report as significant.

Fig 2. Please show scatter plots.

Intensity is misspelled.

Inadequate immunofluorescence microscopy information is provided. What objective was used? what filters? What excitation source? What kind of coverslips?

This is highly relevant for the colocalization experiments. The red patterns look consistently smaller. I'm wondering whether the objective used is corrected for chromatic aberrations. This looks like the images are focused to different focal planes.

Fig. 3f I don't understand the figure. Why are there two lines for each condition and why do they converge at the same endpoint?

Fig. 4. Field is misspelled.

4a. It's a little surprising that the large histogram differences only result in p values in the * category. The use of scatter plots will help readers better understand the data.

Fig 8. The FRET assay is highly problematic. First, the authors claim to have created positive and negative controls, but the data from the controls are not shown. The range of possible FRET values is unclear. A unimolecular positive control is reasonable, but a bimolecular negative control is not. The FRET values are not comparable. Importantly, the reporter construct is in the wrong compartment. TXNDC5 is a resident endoplasmic reticulum protein. TGFBR1 is a protein that would interact with

TXNDC5 in the endoplasmic reticulum. The FRET construct puts fluorescent proteins at each end, which would prevent the signal sequence from targeting the construct into the endoplasmic reticulum. Thus, it should neither fold correctly nor could it interact with TXNDC5. Furthermore, even if the authors created a correctly targeted reporter, they used ECFP and EYFP, which were shown by the Tsien lab back in 2003 in Zacharias et al. (Science) to be prone to dimerization unless the proteins were monomerized with an A206K mutation. This will lead to aberrant positive FRET signals, especially on a bimolecular reporter. Finally, the correct negative control would be an unfoldable mutant TGFBR1, preferably due to a single point mutation. Taken together, the FRET assay is invalid and is unlikely to be correctable without a massive effort, well beyond the scope of this manuscript.

How does TXNDC5 OE influence TGFBR1 levels? In fig 6b, there's nearly a 3 fold increase. In 7b, the value drops to only about a 70% increase (even though the representative blot bands look dramatically more contrasted than a 70% difference). In Fig 8e, the increase with OE is even lower, maybe 40%, and the representative band difference looks more like the presented histograms. This extreme variability to a frankly subtle effect raises serious concerns about the reproducibility of the effect and the proposed mechanism.

I'm confused by the ATF6 assay. BiP levels clearly increase, but ATF6 p50 levels exhibit an extremely modest change in fig 9a. I'm not convinced the antibody actually recognizes the p50 version of the protein. There's only one antibody available that recognizes ATF6 was generated by the Mori lab. ATF6 degrades rapidly and requires special handling for immunoblot assays. Changes in the reported p50 levels are extremely modest. There is also no other standard positive control for ER stress or a time course. BiP levels usually increase hours after UPR induction and ATF6 cleavage is an early event.

The TXNDC5 OE control is not a great choice. Overexpression of any ER protein induces transient ER stress and adaptation. A negative control would either an inactive version of the protein (as used in 8e, though the effects with wt were much more modest in the assay) or an inert resident ER protein.

The luciferase assay is somewhat surprising. The difference increased luciferase production in stressed cells is less than 50%. Luciferase has a tremendous dynamic range of 3 orders of magnitude, and yet message levels are similarly modest in increase in response to stress (less than 50%). In most assays, a 1.5 fold difference is considered the bare minimum for clear evidence of upregulation of message levels. One interpretation is that ATF6 does not directly regulate TXNDC5 levels (a helpful control would be to mutate the proposed ATF6 binding site of the luciferase reporter) and instead, another downstream TF is responsible.

To robustly nail down a mechanism, the authors would need to do immunoprecipitations of wt vs mutant and endogenous TXNDC5 with TGFBR1. As presented, there appears to be something happening, but direct evidence of mechanism is missing.

Reply to Reviewer #1 of the manuscript entitled: "Fibroblast-enriched Endoplasmic Reticulum Protein TXNDC5 Promotes Pulmonary Fibrosis by Augmenting TGF β Signaling through TGFBR1 Stabilization" by Tzu-Han Lee et al, submitted for consideration for publication in Nature Communication (NCOMMS-19-26820).

We thank this Reviewer for the comments and suggestions of the original version of this manuscript. In the paragraphs below, we have responded to comments of this Reviewer and we have indicated, where appropriate, the specific changes made in the manuscript to address the noted concerns.

1. Please make sure that the figure legends for animal studies point out clearly that (eg Figure 2) the tissue extracts and other histology are from day 21 post bleo animals.

We thank this Reviewer for the comments. We apologize for the oversight. We have now provided detailed descriptions about the timing of tissue collection in each of the figure legends related to animal experiments in the revised version of the manuscript.

2. The data in figure 2 show mRNA levels at day 21 of bleo treatment. How does this data compare to the several array data sets in the literature already?? (Kaminski et al 2000; Haston et al 2005 or Cabrera et al 2013).

We thank this Reviewer's questions. We have re-analyzed microarray datasets from mouse lung tissues from Cabrera et al 2013 (GSE42301)¹ and Haston et al 2005 (GSE2640)^{1,2}; however, we could not find the online data set reported in Kaminski's paper. Similar to our findings, data from Cabrera and Haston studies showed that *Txndc5* and multiple fibrogenic genes had a trend toward upregulation in BLM-treated, compared to sham-operated, mouse lungs, most of which, however, did not reach statistical significance owing to limited sample sizes (n=3 for both Sham and BLM in Cabrera study; n=1 for Sham and 2 for BLM in Haston study. See top two panels of the figure shown below). Therefore, we performed additional analysis using another data set with a larger sample size (GSE77326 by Lv X et al), which showed that *Txndc5* and multiple fibrogenic genes (*Col1a1*, *Eln*, *Fn*, *Ctgf*) were significantly upregulated in mouse lungs following BLM treatment on day 21 (n=6), compared to that in Sham controls (n=6, see bottom panel of the figure shown below). Data are presented as mean \pm SEM, *P<0.05, **P<0.01, ***P<0.001, n.s.=non-significant determined using two-tailed unpaired *t* test.

3. Supplementary figures appear to be added to previous concerns about localization of TXNDC5 to fibroblasts only. Endothelial cells appear positive (Supplementary fig 2c and 2d) so what endothelial cell functions or productions are affected in the global KO mice that could affect fibrogenesis??

We thank this Reviewer's questions. Indeed, TXNDC5, also known as EndoPDI, was initially characterized in endothelial cells.³ It is, therefore, not surprising that TXNDC5 was expressed in some endothelial cells of the mouse lung tissues. As endothelial cells have been suggested to contribute to the development of pulmonary fibrosis via endothelial-mesenchymal transition (EndoMT),^{4,5} we cannot exclude the possibility that TXNDC5 could also contribute to pulmonary fibrogenesis through promoting EndoMT in pulmonary endothelial cells. This, however, will require extensive *in vitro* and *in vivo* experiments to confirm and to decipher the underlying molecular mechanisms, which are beyond the scope of this manuscript. In addition, the tamoxifen-inducible, fibroblast-specific conditional knockout mouse line (*Col1a2-Cre/ERT2*Txndc5^{fl/fl}*) used in our study allows efficient deletion of *Txndc5* in collagen-producing, active lung fibroblasts regardless of their origins (i.e. from resident lung fibroblasts, endothelial cells or bone marrow-derived fibrocytes). The conclusion that fibroblast-specific deletion of *Txndc5* significantly lessened the development and progression of pulmonary fibrosis and lung dysfunction induced by BLM in mice, therefore, remains unchanged whether TXNDC5 contributes to EndoMT-mediated lung fibrogenesis or not. We have included the discussion on EndoMT in the revised version of the manuscript (Discussion section, p.17, second paragraph, line 59-72).

4. Figure 3d implicates TXNDC5 in fibroblast accumulation after bleo, but what is the effect of TXNDC5 deletion on other fibroblast functions in other tissues?? (tissue repair, normal growth, female post menstruation restructuring, etc).

We thank this Reviewer's questions. In our previous studies, we have shown that global deletion of *Txndc5* protects against isoproterenol-induced cardiac fibrosis and hypertrophy.⁶ *Txndc5* knockout mice did not show any developmental/growth defects or pathological changes up to 2.5 years of age. They also breed normally, although we do not know for sure if the female post menstruation restructuring is unaffected. Depletion of TXNDC5 in human cardiac fibroblast prevents fibroblasts activation and extracellular matrix productions without causing cellular apoptosis.⁶ Our preliminary data showed that

TXNDC5 deletion also reduced the fibrogenic activities of fibroblasts in the kidneys ⁷ and liver⁸ suggesting that TXNDC5 could be a critical mediator of tissue fibrosis in multiple organs.

5. Fig 3b what does “white areas” refer to? How is it detected? How does this show lung volume reduction?

We thank this Reviewer’s questions. Using the *ex vivo* micro-CT method established by Scotton et al,⁹ we were able to perform 3D analysis of fibrotic lesions and lung volume in the mouse lungs following bleomycin treatment. The white areas in Figure 3b indicate X-ray dense fibrotic lesions that show high grayscale density on the micro-CT sections.⁹ We used the analysis software (CTAn, Bruker microCT, Kontich, Belgium) with X-ray grayscale thresholding and automatic ROI definition to carry out segmentation analysis of the lung tissue, where the volume of fibrotic and non-fibrotic lung areas was quantified for each segment; data for each lung (~900 sections) were then compiled into a composite measurement of fibrotic and non-fibrotic lung volumes. These technical details are now provided in the Methods Section (p. 21, paragraph on *ex vivo* microCT scanning) of the revised version of the manuscript.

6. Also for 6e the details refer to “Targeted deletion of Txndc5” when it is a global deletion and this same usage is in second to last paragraph of discussion “Using global TXNDC5 knockout mouse line we have shown that targeted deletion of TXNDC5 “ can be confusing as to what is the “targeted” cell? In my understanding, targeted implies the gene is being deleted in a specific cell. The experiment to delete in lung fibroblasts (Fig 5 a and b) specifically uses fibroblast in culture and cannot be universally applied or construed to relate to fibroblast specific deletion in vivo.

Very confused aggregation of data in combining individual cell cultures alongside in vivo Bleomycin experiments.

We thank this Reviewer’s comments and we apologize for the confusion. We have replaced “targeted deletion” with “global deletion” to describe the global *Txndc5* knockout mouse line throughout the revised version of the manuscript. We have also rearranged the panels in Fig. 6 as suggested by this Reviewer to improve the data presentation.

7. Fig 6 TXNDC5 knockdown should be shown to be effective by decrease or absence of TXNDC5 mRNA. Again, the use of “targeted deletion” (fig 6d) is confusing and missing the day 21 detail that is needed to clarify the situation in the bleomycin experiment. It was a global deletion mouse that was used.

We thank this Reviewer’s comments and suggestions. We have now included data on *TXNDC5* knockdown efficiency in supplementary Fig. 7, corrected “targeted deletion” with “global deletion”, and provided the day 21 detail in the legends of Fig. 6 in the revised version of the manuscript.

8. The claim is that TXNDC5 facilitates the folding of TGFbR1, but what is the effect of this chaperone on TGFbR2?? Supp figure 7 shows the FRET data for TGFbR1 but what is the data for TGFbR2?? Both are required for proper TGFb1 signalling??

We thank this Reviewer's questions. We have conducted additional experiments to examine whether TXNDC5 interacts with TGFBR2. As shown in Supplementary Fig 8b, proximal ligation assay (PLA) did not reveal measurable interaction between TXNDC5 and TGFBR2. Together with the observation that TXNDC5 depletion/overexpression did not affect TGFBR2 protein expression levels (Figure 6a, b), it is safe to say that TXNDC5 does not interact with or impact the protein expression level of TGFBR2. Although both TGFBR1 and TGFBR2 are both required for proper TGFβ1 signaling, it has been shown that altering TGFBR1 expression level alone is sufficient to strengthen or attenuate TGFβ1 signaling activity.^{10,11} These additional data are now provided in Supplementary Fig 8b and described in the Results section (p. 11, second paragraph, line 93-97) in the revised version of the manuscript.

9. Fig 7b: If TXNDC5 acts through TGFbR1 and not 2 there should be R2 dependent genes that are regulated through TGFb stimulation. You should show this to certify that R1 is the target even though it is recognized that R2 forms a heterodimeric complex with TGF-beta receptor type-1, and binds TGF-beta to phosphorylate proteins, which then enter the nucleus and regulate the transcription of genes related to cell proliferation, cell cycle arrest, wound healing, immunosuppression, and tumorigenesis. Are any or all TGFb responsive genes depleted by KO of R1?

Also co-precipitation is a usual way to show direct protein-protein interaction but Fret analysis seems good for conclusion.

We thank this Reviewer's comments and suggestions. As described by this Reviewer and discussed in the original version of the manuscript, TGFβ1 signaling begins with the binding of TGFβ dimer to the TGFBR2, which recruits, phosphorylates and activates TGFBR1. Activated TGFBR1 then initiates Smad-mediated (canonical) and Smad-independent (non-canonical) signaling pathway, leading to downstream transcriptional regulation. Because the downstream effects of TGFβ1-TGFBR2 binding are mediated exclusively by TGFBR1, most, if not all, TGFBR2-dependent transcriptional regulation should also be dependent on TGFBR1. Consistent with this notion, one study by Larsson et al showed that the transcriptional activation of the CAGA-reporter (a Smad3/Smad4 binding element) by TGFβ1 was completely abrogated in *Tgfbr1*^{-/-} embryonic fibroblasts and endothelial cells,¹² suggesting most of TGFβ1 responsive genes were abolished by knockout of *Tgfbr1*.

We have conducted co-immunoprecipitation experiments and showed the direct protein-protein interaction between TXNDC5 and TGFBR1. These additional data are now provided in Supplementary Fig. 8c in the revised version of the manuscript.

10. For a Nature article in 2019, the background references regarding the myofibroblast and pulmonary fibrosis seem somewhat dated (ref 2,3 and 7). There has been much written about these 2 issues since 2003 and 2013.

We thank this Reviewer's comments and suggestions. We have replaced ref 2, 3 and 7 with more recent references in the background section of the revised version of the manuscript.

11. The most crucial question regarding whether fibroblast specific deletion (therefore fibroblast targeted deletion) of TXNDC5 leads to inhibition of fibrogenesis is addressed in Fig 10 and suggests strongly that this is an important pathway in progressive fibrosis. Critical questions revolve around whether this pathway of intervention in TGFb1 mediated fibrosis is any different from those examined through the use of other ALK 5 inhibitors, (such as LY 364947 or SD201) and can avoid the cardiac toxicity (at least) seen with both of these very powerful TGFb1 inhibitors??

We thank this Reviewer's comments and questions. As this Reviewer pointed out, TGFβ1 signaling plays an essential role in the maintenance of normal cell physiology and homeostasis. Broad inhibition of TGFβ1 signaling, therefore, is known to cause serious side effects including cardiac toxicity. Deletion of TXNDC5, either globally or specifically in fibroblasts, did not lead to discernible adverse effects on cardiac development, structure or function. This could be due to the fact that TXNDC5 is barely expressed in cardiomyocytes,⁶ thereby precluding its direct impact on the TGFβ1 signaling in cardiomyocytes. Consistent with this notion, we have shown that global deletion of *Txndc5* did not lead to any cardiac dysfunction at baseline or following pathological stimulation.⁶ We have discussed the advantage of inhibiting TXNDC5 in the Discussion section of the original and revised version of the manuscript (p.15, first paragraph, line 00-17).

Reply to Reviewer #2 of the manuscript entitled: "Fibroblast-enriched Endoplasmic Reticulum Protein TXNDC5 Promotes Pulmonary Fibrosis by Augmenting TGFβ Signaling through TGFBR1 Stabilization" by Tzu-Han Lee et al, submitted for consideration for publication in Nature Communication (NCOMMS-19-26820).

We thank this Reviewer for the comments and suggestions of the original version of this manuscript. In the paragraphs below, we have responded to comments of this Reviewer and we have indicated, where appropriate, the specific changes made in the manuscript to address the noted concerns.

The study by Lee et al seeks to understand the pathogenic mechanisms underlying the progression of pulmonary fibrosis. The study's major focus is on the protein TXNDC5, a disulfide isomerase abundantly expressed in the ER of mammalian cells. Interest in this protein relates to a recent report by this group showing that deficiency of TXNDC5 exacerbates cardiac fibrosis in rodents. In this study, the focus is on TXNDC5 expression in the lung. The major findings are that levels of TXNDC5 are

increased in lungs of IPF patients as well as bleomycin-exposed mouse lungs, and that global or fibroblast-specific deficiency of *TXNDC5* ameliorates experimentally induced pulmonary fibrosis in mice. Mechanistically, investigators also purport that *TXNDC5* is critical for regulating myofibroblast differentiation by controlling the expression of TGFβ receptor 1 levels. Overall, findings in this publication are likely to be of interest to the wide readership audience of this journal, although conclusions are somewhat oversimplified and seem to argue against the current paradigm of ER stress playing a pathogenic role in fibrotic lung diseases. To support the conclusions of the manuscript the following studies should be considered.

Major

1) The mechanistic connection between *TXNDC5* and ER stress is rather weak. Please examine more than one ER stress pathway, namely PERK and IRE and their downstream effectors as well as consider verifying that protein aggregation is increased in cells.

We thank this Reviewer's comments and suggestions. We have conducted additional experiments to examine PERK and IRE1 pathways and the link between *TXNDC5* and ER stress. As shown in Figure 9a, b and Supplementary Fig. 9, TGFβ1 treatment in human pulmonary fibroblasts (HPF) led to marked activation of components of all three branches of ER stress pathways, including ATF6, XBP-1s (IRE1 pathway) and ATF4 (PERK pathway). With the concomitant treatment of ER stress inhibitor 4-PBA, the activation of ER stress markers BiP and ATF6α(N) by TGFβ1 were blocked, and TGFβ1-induced transcriptional upregulation of *TXNDC5* was markedly attenuated. These data suggest that increased ER stress level is required for TGFβ1-induced *TXNDC5* upregulation. Further experiments showed that ATF6, but not XBP1 (IRE1 pathway) or EIF2A (PERK pathway), is required for TGFβ1-induced *TXNDC5* transcriptional activity (Figure 9 c,d,e and Supplementary Fig. 10). Together, these data indicate that *TXNDC5* is upregulated by TGFβ1, the effects of which are abrogated by inhibition of ER stress, specifically ATF6 pathway. These results are described in the Results section (p. 12, second paragraph) in the revised version of the manuscript.

2) As a control, please knockdown either the PERK or IRE pathways (regardless of whether elevated) to confirm that *TXNDC5* is not regulated through other arms of the unfolded protein response. Otherwise, it is somewhat premature to conclude that everything is regulated through ATF6.

We thank this Reviewer's comments and suggestions. We have conducted additional experiments to knockdown components of PERK and IRE1 pathway in HPF, with or without TGFβ1 stimulation. As shown in Supplementary Fig. 10, knocking down XBP1 (IRE1 pathway) or EIF2A (PERK pathway) did not affect TGFβ1-induced *TXNDC5* upregulation. Together with the observation that ATF6 depletion abrogates TGFβ1-induced *TXNDC5* upregulation (Figure 9d), these data suggest that TGFβ1 trigger *TXNDC5* expression through ATF6-mediated transcriptional control. These data are now described in the Results section (p. 12, second paragraph) in the revised version of the manuscript.

3) The idea of knocking down TXNDC5 to reduce pulmonary fibrosis is somewhat confusing since one would expect that ER stress would be increased in the absence of this proteins expression (cells would have a harder time folding proteins). To address this, please examine markers of ER stress in lung and fibroblasts of TXNDC5 expressing and non-expressing cells.

We thank this Reviewer's comments and suggestions. We have conducted additional experiments to examine ER stress markers in the lung tissues and primary lung fibroblasts from WT and *Txndc5*^{-/-} mice. As shown in Supplementary Fig. 14 a-d, loss of TXNDC5 did not lead to measurable changes in ER stress markers such as BiP, XBP-1 and ATF4, in the mouse lung tissue and isolated mouse lung fibroblasts. Because the protein disulfide isomerase (PDI) family consists of 21 members, it is possible that functional redundancy exists between TXNDC5 and other PDI family proteins, thereby preventing overt increases in ER stress in the absence of TXNDC5. These data have been described in the Discussion section (p. 16, first paragraph, line 32-34) in the revised version of the manuscript.

4) Figure 2 B and C do not permit adequate visualization of TXNDC5 expression in type II cells, which are also known to upregulate UPR machinery in response to bleomycin. Please provide low power images of lung under bleomycin conditions to confirm that TXNDC5 is not expressed in these cells.

We thank this Reviewer's comments and suggestions. We have conducted additional experiments to determine if TXNDC5 is expressed in type II cells in response to bleomycin. IF staining of the lung sections from BLM-treated *Spc-Cre/ERT2*ROSA26-mTmG* mice (*SPC-mTmG*), which allow visualization of type II pneumocytes with GFP, was performed and showed that TXNDC5 is not expressed at appreciable levels in type II pneumocytes (Supplementary Fig. 2g). Consistent with this observation, re-analysis of a microarray dataset (GSE81067) obtained from human normal lung fibroblasts (HFL-1) and type II epithelial cells (A549) showed that the expression level of *TXNDC5* mRNA was much lower in in type II pneumocytes than in lung fibroblasts (figure shown below). These data have been described in the Results section (p. 7, first paragraph, line 74-80) in the revised version of the manuscript.

(Data are presented as mean ± SEM, ***P<0.001 determined using two-tailed unpaired *t* test)

5) Please confirm that TXNDC5 deficiency fibroblast are still viable. Otherwise, it is possible that increased cell death explains the reduction in myofibroblast markers after knockdown of this protein. We thank this Reviewer's comments and suggestions. We have performed additional experiments to determine the viability of lung fibroblasts with TXNDC5 deficiency. As shown in Supplementary Fig. 14e, flow-cytometry analysis did not show significant differences in the percentages of positive propidium iodide (PI) and annexin V staining of primary mouse lung fibroblasts (MLF) isolated from WT and *Txndc5*^{-/-} mice, suggesting that TXNDC5 deficiency did not lead to increased apoptosis in lung fibroblasts. These data have been described in the Discussion section (p. 16, first paragraph, line 32-34) in the revised version of the manuscript.

Reply to Reviewer #3 of the manuscript entitled: "Fibroblast-enriched Endoplasmic Reticulum Protein TXNDC5 Promotes Pulmonary Fibrosis by Augmenting TGF β Signaling through TGFBR1 Stabilization" by Tzu-Han Lee et al, submitted for consideration for publication in Nature Communication (NCOMMS-19-26820).

We thank this Reviewer for the comments and suggestions of the original version of this manuscript. In the paragraphs below, we have responded to comments of this Reviewer and we have indicated, where appropriate, the specific changes made in the manuscript to address the noted concerns.

The authors have performed a number of assays to report a connection between expression of a protein disulfide isomerase and pulmonary fibrosis. There is an interesting story here. However, there are some issues with internal consistency and the proposed mechanism. There's clearly an effect on lung tissue and cells. Whether it is related to TGFBR1 is debatable.

My main concerns are to change the histograms to scatter plots to more accurately present the data, provide more details about some of the imaging assays, reconcile differences between experiments, and to get rid of the deeply flawed FRET assay. Details are listed below.

Fig. 1. Rather than cover up the noisiness of the data with block histograms, the authors should present scatter plots that show the range of the data. Please state whether errors are reported as SD or SEM.

Fig. 1b. I disagree with the conclusion that there is a strong correlation. Collagen levels are highly variable relative to TXNDC5 levels. The R value reflects an averaging effect, not a good fit.

Fig 1e. FN data inclusion is questionable. There does appear to be a difference between low and high TXNDC5 expression and FN levels, though the FN level differences are quite modest, maybe 30% at best. That seems too low to report as significant.

We thank this Reviewer's comments and suggestions. In the revised version of the manuscript, we have

changed all the bar graphs to scatter plots with errors reported as SEM. In addition, we have removed figure 1b and FN data from figure 1e (now figure 1d) as suggested by this Reviewer.

Fig 2. Please show scatter plots.

Intensity is misspelled.

Inadequate immunofluorescence microscopy information is provided. What objective was used? what filters? What excitation source? What kind of coverslips?

This is highly relevant for the colocalization experiments. The red patterns look consistently smaller. I'm wondering whether the objective used is corrected for chromatic aberrations. This looks like the images are focused to different focal planes.

We thank this Reviewer's comments and questions. We have changed the bar graphs to scatter plots and corrected the misspelling in fig. 2b. We apologize for not providing the detailed information of the immunofluorescence microscopy used in this study. The objective was Olympus UPLFLN 40X Objective without chromatic aberrations correction, and the filters were U-MWB2 (FITC) (Ex/Em:460-490/520IF (nm)), U-MWG2 (TRITC) (Ex/Em:510-550/590 (nm)), and U-MWU2 (DAPI) (Ex/Em:330-385/420 (nm)). The excitation source was a mercury lamp. The coverslips were from Deckglaser Microscope Cover Glass, and the thickness was 0.13-0.16mm. We have provided the detailed information of immunofluorescence microscopy in the Methods section (p. 23, second paragraph) in the revised version of the manuscript. In Figure 2c, COL1A1-GFP is expressed in the entire cytoplasm of active, collagen-secreting fibroblasts, whereas TXNDC5 is distributed mainly in the perinuclear ER in these cells. The size of TXNDC5-positive area (red color), therefore, would be smaller than COL1A1-GFP-positive area (green color).

Fig. 3f I don't understand the figure. Why are there two lines for each condition and why do they converge at the same endpoint?

We thank this Reviewer's questions. These pressure-volume (PV) curves can be executed in a pressure-driven manner. The volume at zero pressure can only be obtained by extrapolating, as a measurement at zero pressure is only possible if the lungs are fully degassed. Therefore, the inflation typically starts at the set PEEP (positive end-expiratory pressure) value. From there, the subject's lungs are inflated to a default pressure of 30 cmH₂O in a ramp style manner, and then deflated back to the initial pressure. The inflation and deflation arms differ at any given pressure because of the hysteresis of the lungs, which is attributable to the viscoelastic and recoil properties of the lung tissue as well as to different surface tension forces during inflation and deflation. Therefore, there are two lines for each condition. Figure here shows the result from the "Guide for flexiVent Techniques & Measurements". (Website: [https://www.scireq.com/flexivent/techniques-and-measurements/.](https://www.scireq.com/flexivent/techniques-and-measurements/))

Fig. 4. Field is misspelled.

4a. It's a little surprising that the large histogram differences only result in p values in the * category. The use of scatter plots will help readers better understand the data.

We thank this Reviewer's suggestions. We have corrected the misspelling and changed the bar graphs to scatter plots in the revised version of the manuscript.

Fig 8. The FRET assay is highly problematic. First, the authors claim to have created positive and negative controls, but the data from the controls are not shown. The range of possible FRET values is unclear. A unimolecular positive control is reasonable, but a bimolecular negative control is not. The FRET values are not comparable. Importantly, the reporter construct is in the wrong compartment. TXNDC5 is a resident endoplasmic reticulum protein. TGFBR1 is a protein that would interact with TXNDC5 in the endoplasmic reticulum. The FRET construct puts fluorescent proteins at each end, which would prevent the signal sequence from targeting the construct into the endoplasmic reticulum. Thus, it should neither fold correctly nor could it interact with TXNDC5. Furthermore, even if the authors created a correctly targeted reporter, they used ECFP and EYFP, which were shown by the Tsien lab back in 2003 in Zacharias et al. (Science) to be prone to dimerization unless the proteins were

monomerized with an A206K mutation. This will lead to aberrant positive FRET signals, especially on a bimolecular reporter. Finally, the correct negative control would be an unfoldable mutant TGFBR1, preferably due to a single point mutation. Taken together, the FRET assay is invalid and is unlikely to be correctable without a massive effort, well beyond the scope of this manuscript.

We thank this Reviewer's comments and suggestions. These are very important points and we agree with this Reviewer's assessment on the FRET experiments. We have removed the FRET data in the revised version of the manuscript.

How does TXNDC5 OE influence TGFBR1 levels? In fig 6b, there's nearly a 3 fold increase. In 7b, the

value drops to only about a 70% increase (even though the representative blot bands look dramatically more contrasted than a 70% difference). In Fig 8e, the increase with OE is even lower, maybe 40%, and the representative band difference looks more like the presented histograms. This extreme variability to a frankly subtle effect raises serious concerns about the reproducibility of the effect and the proposed mechanism.

We thank this Reviewer's comments and questions. We agree with this Reviewer's observation that variability exists in the extent of TGFBR1 upregulation in response to TXNDC5 OE in figure 6b, 7b and 8e. The extent of TGFBR1 upregulation was 2.8-fold, 1.7-fold and 1.5-fold, comparing to empty vector controls, in figure 6b, 7b and 8e, respectively. The variability of these results, however, could be due to the differences in the magnitude of TXNDC5 OE in each of the experiments. In Fig. 6b, TXNDC5 was overexpressed mostly at high 3- to low 4-fold, whereas TXNDC5 was largely overexpressed at low 3-fold and ~2-fold in Fig. 7b and 8e, respectively. Higher TXNDC5 OE levels could lead to greater extent of TGFBR1 upregulation, although the effects may not be linear. TXNDC5 OE, though with variability, consistently resulted in TGFBR1 upregulation in HPF, which was in sharp contrast to TGFBR2, the level of which remained unchanged irrespective of the extent of TXNDC5 OE.

I'm confused by the ATF6 assay. BiP levels clearly increase, but ATF6 p50 levels exhibit an extremely modest change in fig 9a. I'm not convinced the antibody actually recognizes the p50 version of the protein. There's only one antibody available that recognizes ATF6 was generated by the Mori lab. ATF6 degrades rapidly and requires special handling for immunoblot assays. Changes in the reported p50 levels are extremely modest. There is also no other standard positive control for ER stress or a time course. BiP levels usually increase hours after UPR induction and ATF6 cleavage is an early event.

We thank this Reviewer's comments and suggestions. We have conducted additional immunoblots using the ATF6 antibody generated by the Mori lab as suggested by this Reviewer (Fig. 9a and Supplementary Fig. 9a,b). In addition, we included tunicamycin, a well-accepted ER stress inducer, -treated HPF samples as positive controls (Supplementary Fig. 9a, b). These data are now provided in the Results section (p. 12, second paragraph) in the revised version of the manuscript.

The TXNDC5 OE control is not a great choice. Overexpression of any ER protein induces transient ER stress and adaptation. A negative control would either an inactive version of the protein (as used in 8e, though the effects with wt were much more modest in the assay) or an inert resident ER protein.

We thank this Reviewer's comments. Based on the context of the comments, we assume this Reviewer was referring to Fig. 9b in the original version of the manuscript. We apologize for not labeling/describing the data in Fig. 9b more clearly. In Fig 9b, *TXNDC5* was induced by TGFβ1 treatment, and the upregulation of *TXNDC5* by TGFβ1 treatment could be prevented by the co-treatment of ER

stress inhibitor 4-PBA. TXNDC5 expression vectors were NOT used in these experiments. We have modified the labeling of Fig. 9b in the revised version of the manuscript.

The luciferase assay is somewhat surprising. The difference increased luciferase production in stressed cells is less than 50%. Luciferase has a tremendous dynamic range of 3 orders of magnitude, and yet message levels are similarly modest in increase in response to stress (less than 50%). In most assays, a 1.5 fold difference is considered the bare minimum for clear evidence of upregulation of message levels. One interpretation is that ATF6 does not directly regulate TXNDC5 levels (a helpful control would be to mutate the proposed ATF6 binding site of the luciferase reporter) and instead, another downstream TF is responsible.

We thank this Reviewer's comments and suggestions. In our experiments (Fig. 5b, 9b and 9d), TGF β 1 (10 ng/ml) treatment consistently led to significant upregulation of *TXNDC5* transcripts in HPF, ranging from 1.4-2.1 fold, the extent of which was similar to that of *TXNDC5* protein/transcript upregulation in the lung tissues and lung fibroblasts from human IPF patients (Fig. 1a-c) and mouse lungs following BLM treatment (Fig. 2a, 6c and Supplementary Fig. 5a and 12b). We have repeated the luciferase reporter assays (this time with co-transfected Renilla luciferase vector to control for transfection efficiency) and the results showed that TGF β 1 treatment led to an average of ~1.8-fold increase in the transcriptional activity of WT *TXNDC5* promoter, whereas the mutant *TXNDC5* promoter reporter construct lacking the ATF6 binding site (TGACGTGG, +642 to +653, Δ ATF6, Fig. 9e) failed to respond to TGF β 1 stimulation. Taken together, the *in vitro/ in vivo* experiments and luciferase reporter assays demonstrate a consistent transcriptional upregulation of *TXNDC5* in response to fibrogenic stimuli, which is, at least in part, dependent on ATF6-mediated transcriptional regulation.

To robustly nail down a mechanism, the authors would need to do immunoprecipitations of wt vs mutant and endogenous TXNDC5 with TGFBR1. As presented, there appears to be something happening, but direct evidence of mechanism is missing.

We thank this Reviewer's comments and suggestions. We have conducted additional experiment to examine the interaction between TGFBR1 and WT (or mutant) *TXNDC5*. As shown in Supplementary Fig. 8c and d, co-immunoprecipitation experiments showed direct binding between TGFBR1 and WT, but not AAA mutant, *TXNDC5*, suggesting that *TXNDC5* interacts with TGFBR1 via its functional domains that mediates its protein disulfide isomerase activity. These data are now described in the Results section (p11, third paragraph, line 05-07) in the revised version of the manuscript.

References

1. Cabrera, S., *et al.* Gene expression profiles reveal molecular mechanisms involved in the

progression and resolution of bleomycin-induced lung fibrosis. *Am J Physiol Lung Cell Mol Physiol* **304**, L593-601 (2013).

2. Haston, C.K., Tomko, T.G., Godin, N., Kerckhoff, L. & Hallett, M.T. Murine candidate bleomycin induced pulmonary fibrosis susceptibility genes identified by gene expression and sequence analysis of linkage regions. *J Med Genet* **42**, 464-473 (2005).
3. Sullivan, D.C., *et al.* EndoPDI, a novel protein-disulfide isomerase-like protein that is preferentially expressed in endothelial cells acts as a stress survival factor. *J Biol Chem* **278**, 47079-47088 (2003).
4. Hashimoto, N., *et al.* Endothelial-mesenchymal transition in bleomycin-induced pulmonary fibrosis. *Am J Respir Cell Mol Biol* **43**, 161-172 (2010).
5. Choi, S.H., *et al.* A Hypoxia-Induced Vascular Endothelial-to-Mesenchymal Transition in Development of Radiation-Induced Pulmonary Fibrosis. *Clin Cancer Res* **21**, 3716-3726 (2015).
6. Shih, Y.C., *et al.* Endoplasmic Reticulum Protein TXNDC5 Augments Myocardial Fibrosis by Facilitating Extracellular Matrix Protein Folding and Redox-Sensitive Cardiac Fibroblast Activation. *Circ Res* **122**, 1052-1068 (2018).
7. Chen, Y.-T., Jhao, P.-Y., Lin, S.-L., Hung, C.-T. & Yang, K.-C. FP068ENDOPLASMIC RETICULUM PROTEIN TXNDC5 CONTRIBUTES TO RENAL FIBROGENESIS IN CHRONIC KIDNEY DISEASES. *Nephrology Dialysis Transplantation* **33**, i71-i71 (2018).
8. CT Hung, Y.C., TH Su, SL Lin, KC Yang*. Endoplasmic Reticulum Protein TXNDC5 Promotes Liver Fibrosis by Activating Hepatic Stellate Cells via SMAD-Independent Transforming Growth Factor-beta Signaling. in *American Association for the Study of Liver Diseases 2018, Abstract number 1112*. (2018).
9. Scotton, C.J., *et al.* Ex vivo micro-computed tomography analysis of bleomycin-induced lung fibrosis for preclinical drug evaluation. *Eur Respir J* **42**, 1633-1645 (2013).
10. Zhao, Y., *et al.* USP2a Supports Metastasis by Tuning TGF-beta Signaling. *Cell Rep* **22**, 2442-2454 (2018).
11. Kim, W., *et al.* TFAP2C-mediated upregulation of TGFBR1 promotes lung tumorigenesis and epithelial-mesenchymal transition. *Exp Mol Med* **48**, e273 (2016).
12. Larsson, J., *et al.* Abnormal angiogenesis but intact hematopoietic potential in TGF-beta type I receptor-deficient mice. *EMBO J* **20**, 1663-1673 (2001).

REVIEWERS' COMMENTS:

Reviewer #2 (Remarks to the Author):

Authors have addressed all previous criticisms.

Reviewer #3 (Remarks to the Author):

The authors made a good faith effort to address most of my requests. I recommend acceptance of the revised manuscript.

I would encourage the authors in future studies to use more robust metrics of UPR activation. BiP levels are great, but xbp1 splicing and IRE1 and PERK phosphorylation are better than antibodies against spliced xbp1 protein or downstream targets of eIF2a phosphorylation that are not specific to just PERK activation.